# Mirtron-mediated RNA knockdown/replacement therapy for the treatment of dominant retinitis pigmentosa

Harry O. Orlans [1,2✉], Michelle E. McClements [1], Alun R. Barnard [1],
Cristina Martinez-Fernandez de la Camara[1] & Robert E. MacLaren [1,2]

Rhodopsin (RHO) gene mutations are a common cause of autosomal dominant retinitis pigmentosa (ADRP). The need to suppress toxic protein expression together with mutational heterogeneity pose challenges for treatment development. Mirtrons are atypical RNA interference effectors that are spliced from transcripts as short introns. Here, we develop a novel mirtron-based knockdown/replacement gene therapy for the mutation-independent treatment of RHO-related ADRP, and demonstrate efficacy in a relevant mammalian model. Splicing and potency of rhodopsin-targeting candidate mirtrons are initially determined, and a mirtron-resistant codon-modified version of the rhodopsin coding sequence is validated in vitro. These elements are then combined within a single adeno-associated virus (AAV) and delivered subretinally in a Rho$^{P23H}$ knock-in mouse model of ADRP. This results in significant mouse-to-human rhodopsin RNA replacement and is associated with a slowing of retinal degeneration. This provides proof of principle that synthetic mirtrons delivered by AAV are capable of reducing disease severity in vivo.

[1] Nuffield Laboratory of Ophthalmology, Level 6, West Wing, John Radcliffe Hospital, Headley Way, Headington, Oxford, UK. [2] Moorfields Eye Hospital, London, UK. ✉email: Enquiries@eye.ox.ac.uk

nherited retinal diseases (IRDs) such as retinitis pigmentosa (RP) are the most common cause of irreversible sight loss in the young[1]. The development of gene replacement therapy using adeno-associated viral (AAV) vectors has led to the market approval of Luxturna for the treatment of Leber congenital amaurosis type 2 (LCA2)[2] and AAV-based therapeutics for other IRDs are showing promise in early-phase clinical trials[3–5]. Mutations in the rhodopsin gene (*RHO*) are frequently responsible for autosomal dominant (AD) RP[6,7]. This gene encodes a seven-transmembrane receptor that is expressed at high levels in outer segments (OS) of rod photoreceptors of the retina[8]. In conjunction with the chromophore 11-cis retinal, rhodopsin protein is responsible for initiating the rod photo-transduction cascade. In rhodopsin-related RP, rods degenerate after which cone cells are lost by secondary mechanisms. The mechanisms through which mutant rhodopsin proteins cause rod cell dysfunction and death vary by mutation, but may be broadly categorised into those that have dominant negative effects and those that induce direct cellular toxicity[9]. In contrast to gene therapy for recessive diseases such as LCA2, where simple gene augmentation may be sufficient, treatment of dominant diseases may require suppression of a toxic mutant protein[10].

RNA interference (RNAi) is an important naturally-occurring mode of eukaryotic gene regulation whereby short single-stranded RNAs target specific transcripts for degradation through base pair complementarity[11]. Canonical effectors of RNAi known as micro (mi)RNAs originate from hairpin-like structures often found within introns. These are cleaved from genomically-derived mRNAs at their 5′ end by a microprocessing complex consisting of the enzyme Drosha and its cofactor DGCR8[12,13]. RNAi has been applied to gene therapy as a method of gene silencing, in particular for the treatment of dominantly-inherited diseases[14–16]. Artificial miRNAs driven by RNA polymerase II (Pol-II) promoters[17–19] as well as short hairpin (sh) RNAs driven by RNA polymerase III (Pol-III) promoters[14–16] have been developed for this purpose. Mutational heterogeneity represents a significant challenge for application of RNAi for dominant diseases such as rhodopsin-related ADRP, suggesting the requirement for sequence-specific effectors for each point mutation within a gene. An innovative solution is the so-called knockdown/replacement paradigm. Here both wild-type and mutant alleles are indiscriminately targeted for suppression by a single effector. At the same time, a version of the wild-type gene modified to render it resistant to this effector, most commonly through use of synonymous codons at the target site, is co-delivered. For the case of RNAi, this mutation-independent approach constitutes RNA replacement and has been successfully applied to animal models of dominant disease[14–16].

Several key barriers however exist for the translation of RNAi-based knockdown/replacement gene therapies to clinical trial. Firstly, the expression of the two arms of the therapy within target cells must be carefully balanced. Indeed, Pol-III-driven shRNA may cause cellular toxicity through saturation of endogenous processing systems[20], whilst transgene overexpression, particularly in the case of rhodopsin, may be similarly toxic[21–23]. Secondly, off-target effects must be minimised. The 5′ cleavage site within RNAi hairpins by Drosha/DGCR8 is notoriously variable[24]. The result is a mixed population of mature miRNAs with variations present at the critical 5′ seed region, which risks unanticipated suppression of unintended transcripts. Thirdly, the possibility of off-target effects in bystander cell populations- such as cones and the retinal pigment epithelium (RPE) for IRD treatment- must be considered. Delivery of shRNAs under control of Pol-III promoters such as H1 and U6 has been commonly used for knockdown/replacement therapy[14–16], but as these promoters lack cellular specificity, significant expression in non-targeted cells would be expected.

One novel approach that addresses these concerns uses artificial mirtrons as effectors of RNAi[25]. Mirtrons form an atypical sub-class of naturally-occurring pre-miRNA. These hairpin-like structures are spliced from pre-mRNAs as short introns. Unlike canonical miRNAs, their often critical 5′ end is precisely defined by the splicing machinery. Mirtron-RNAi guide strand generation is thus Drosha/DGCR8-independent, resulting in greater accuracy and avoiding a potential molecular bottleneck[26]. Since mirtrons may be expressed by cell-specific Pol-II promoters, overexpression-related toxicity is unlikely[17,27], and off-target effects in unintended cell populations are minimised. Further, as mirtrons may be expressed bicistronically with a transgene by a common promoter, very precise spatiotemporal matching of the two arms of a knockdown/replacement gene therapy may be achieved.

Gene silencing using artificial mirtrons has been shown in cell culture systems[28–30] and mirtron-mediated knockdown/replacement of the ataxin-7 gene has been demonstrated in vitro[31]. Achieving a construct capable of efficient knockdown together with adequate gene supplementation, an essential requirement for the treatment of dominant disease, has been a significant challenge[31]. Furthermore, efficacy of artificial mirtrons in vivo has not been previously demonstrated.

In this study we have designed and validated artificial mirtrons directed against rhodopsin, and developed a cell-specific expression system for knockdown/replacement gene therapy application that provides significant target gene knockdown without compromise of transgene expression. We show that this expression cassette can drive RNA-replacement when delivered to the mammalian retina by AAV in vivo, and demonstrate an associated rescue effect in a relevant mouse model of rhodopsin-related ADRP.

## Results

**Splicing efficiency and potency of artificial mirtrons in vitro.** Seven candidate mirtrons (M1–M7) were designed to target both mouse and human rhodopsin transcripts so a vector validated in the former species may be suitable for clinical trial. Two versions of M5 differing by two nucleotides were established to account for differences in the human/mouse target sequence (designated M5$^H$ and M5$^M$ respectively). Design methodology and in silico analysis are described within the methods. To determine splicing efficiency, each mirtron was individually cloned into the coding sequence (CDS) of enhanced green fluorescent protein (*eGFP*) as an artificial intron (M1-GFP to M7-GFP plasmids; Fig. 1a). Failure to accurately splice should abolish eGFP expression so that eGFP-derived fluorescence reflects splicing efficiency of nested mirtrons. Following transfection of HEK293 cells, fluorescent signal was detected in all cases but strength varied significantly for the seven mirtron designs (Fig. 1b, c).

To investigate splice variants, cDNA was derived from transfected cells and a set of mirtron-spanning primers directed against the flanking *eGFP* sequence was used to amplify the spliced region. Resulting amplicons corresponded to various splice products: a 225 bp band represented accurate splicing, a 301 bp band represented unspliced mRNA, whilst intermediate bands indicated mis-spliced transcripts resulting from cryptic splice donor/acceptor sites within mirtrons (Fig. 1d, e). These data confirm that rhodopsin-targeting synthetic mirtrons may be spliced as artificial introns in vitro, although efficiency and accuracy of this process is sequence-dependent.

Potency of candidate mirtrons directed against human/mouse rhodopsin was determined in vitro using a dual luciferase assay (Dual Glo®, Promega, UK). Significant knockdown was observed for M2 and M3, and for each version of M5 against its

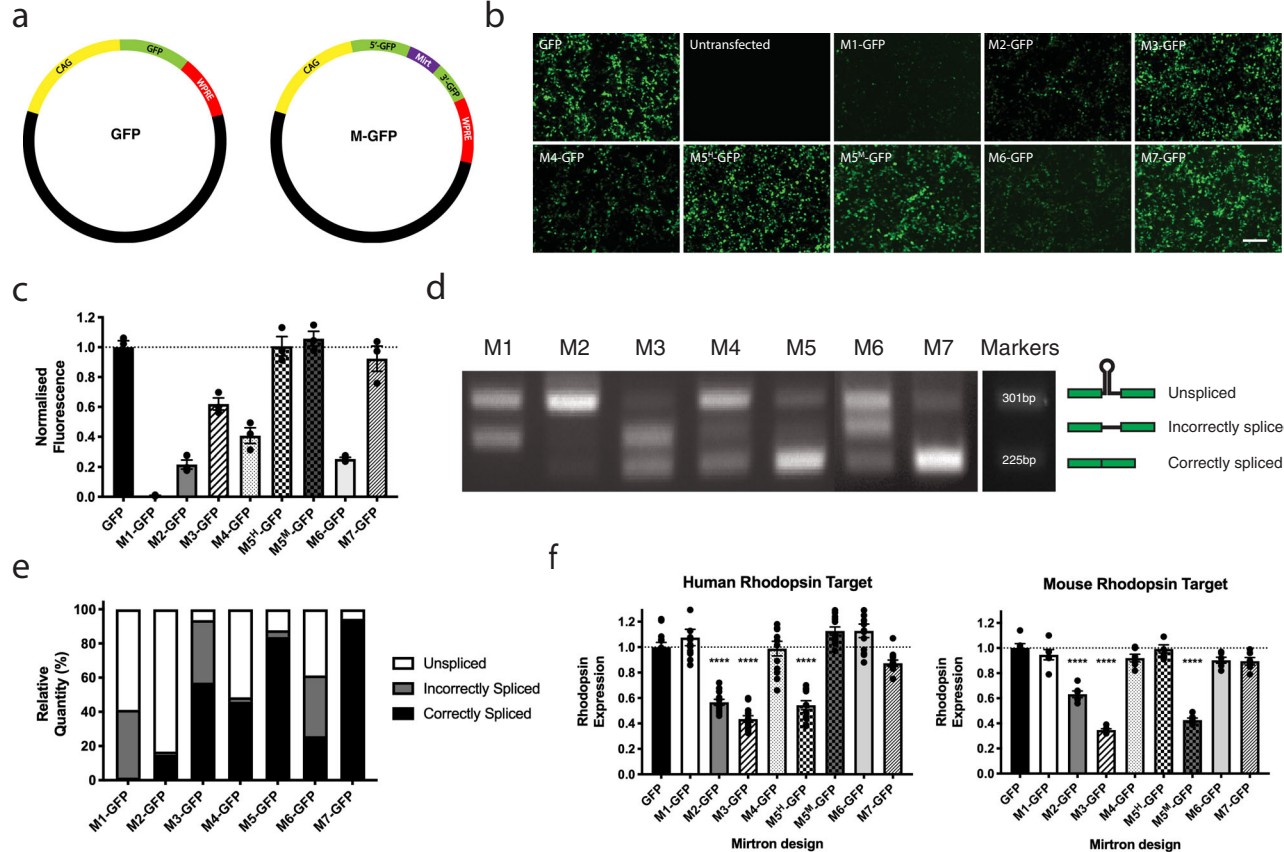

**Fig. 1 Splicing efficiency and in vitro potency of candidate mirtron designs. a** GFP and Mirtron-GFP plasmid maps. The 76 bp mirtron is nested within the *eGFP* CDS dividing it into 5′ and 3′ fragments. **b** Fluorescence micrographs of HEK293 cells taken 48 h post-transfection with GFP and M-GFP plasmids using identical acquisition settings. Scale bar = 200 μm. Experiment repeated three times. **c** Fluorescence levels (mean ± SEM) of protein lysates from M-GFP transfected HEK293 samples normalised to that of the GFP-transfected control group ($n = 3$ each). **d** Representative UV agarose gel image of splice PCR products. The schematic to the right indicates the predicted origin of bands detected. Experiment repeated three times. **e** Relative densitometry of bands corresponding to PCR products derived from a mirtron-spanning primer pair (shown in **d**). Note that proportions of correctly spliced product as determined by this method (black bars) broadly reflect corresponding estimates of splicing efficiency derived from the fluorescence assay (shown in **c**). **f** Determination of mirtron-mediated rhodopsin suppression for human ($n = 12$) and mouse ($n = 6$) rhodopsin targets by the Dual Glo® luciferase assay (Promega, UK). Mean ± SEM is plotted in all cases. Note that each version of M5 was only effective at targeting the rhodopsin sequence of its corresponding species. Ordinary one-way ANOVA for effect of mirtron type on relative luciferase ratio for human target $F_{(8, 99)} = 41.9$, $p < 0.0001$; and for mouse target $F_{(8, 45)} = 77.88$, $p < 0.0001$; ****$p < 0.0001$ versus the mirtronless CAG.eGFP.WPRE plasmid, Dunnett's multiple comparisons test.

corresponding rhodopsin species. No knockdown was apparent for M5$^H$ against mouse rhodopsin, or for M5$^M$ against human rhodopsin, suggesting intolerance of guide strand-target mismatches (Fig. 1f). The effect of the most potent mirtron (M3) against selected putative off-targets was explored with no activity detected (Supplementary note 1).

**Delivery of artificial mirtrons within the 5′-UTR of a transgene.** Placement of mirtrons within the CDS for knockdown/replacement gene therapy may adversely affect transgene expression[31], whilst placement within the 3′-UTR might similarly result in nonsense-mediated decay (NMD)[32,33]. The mechanism of NMD is suggested to rely upon detection of exon-exon junctional complexes (EJCs). These protein complexes are deposited 20–24 nucleotides upstream of 5′ splice sites after splicing, and are usually removed by ribosomes during translation. Introns downstream of a stop codon will not have their EJCs removed in this way as the ribosome has already separated from the transcript. The presence of retained EJCs on mRNA is thought to be the trigger for NMD. Indeed, a premature stop codon induces NMD of the transcript in this manner[32,33]. On this basis, we reasoned that placing mirtrons downstream of the stop codon of our transgene (i.e. in the 3′-UTR)

might trigger degradation of the whole transcript. We therefore explored placing the most effective mirtrons within the 5′-UTR of a transgene (eGFP) in vitro (Fig. 2a). All were more potent in this configuration than when nested within the CDS for both human and mouse rhodopsin targets (Fig. 2b) PCR splice analysis indicated a significantly greater proportion of correctly spliced transcripts for 5′-UTR than for nested mirtrons. Sanger sequencing of the smallest band confirmed that this corresponded to correctly-spliced mRNA (Fig. 2c–e). When multiple mirtrons were included in tandem within the 5′-UTR of *eGFP* (either two copies of M3 or one copy each of M3 and M5), both appeared to splice independently (Fig. 2f, g), and their rhodopsin-suppressing effects were additive (Fig. 2h–k). The effect of 5′-UTR mirtrons on downstream gene expression was determined by fluorescence spectroscopy using lysates derived from transfected HEK293 cells. A single mirtron within the 5′-UTR had no effect on transgene expression whilst a modest reduction in fluorescence was observed when two mirtrons were included (Supplementary Fig. 2).

**Assessment of codon-modified human rhodopsin.** For an effective knockdown/replacement vector, the *RHO* within the AAV must be resistant to co-expressed mirtrons. To achieve this, codons constituting the target sites for the most effective mirtrons

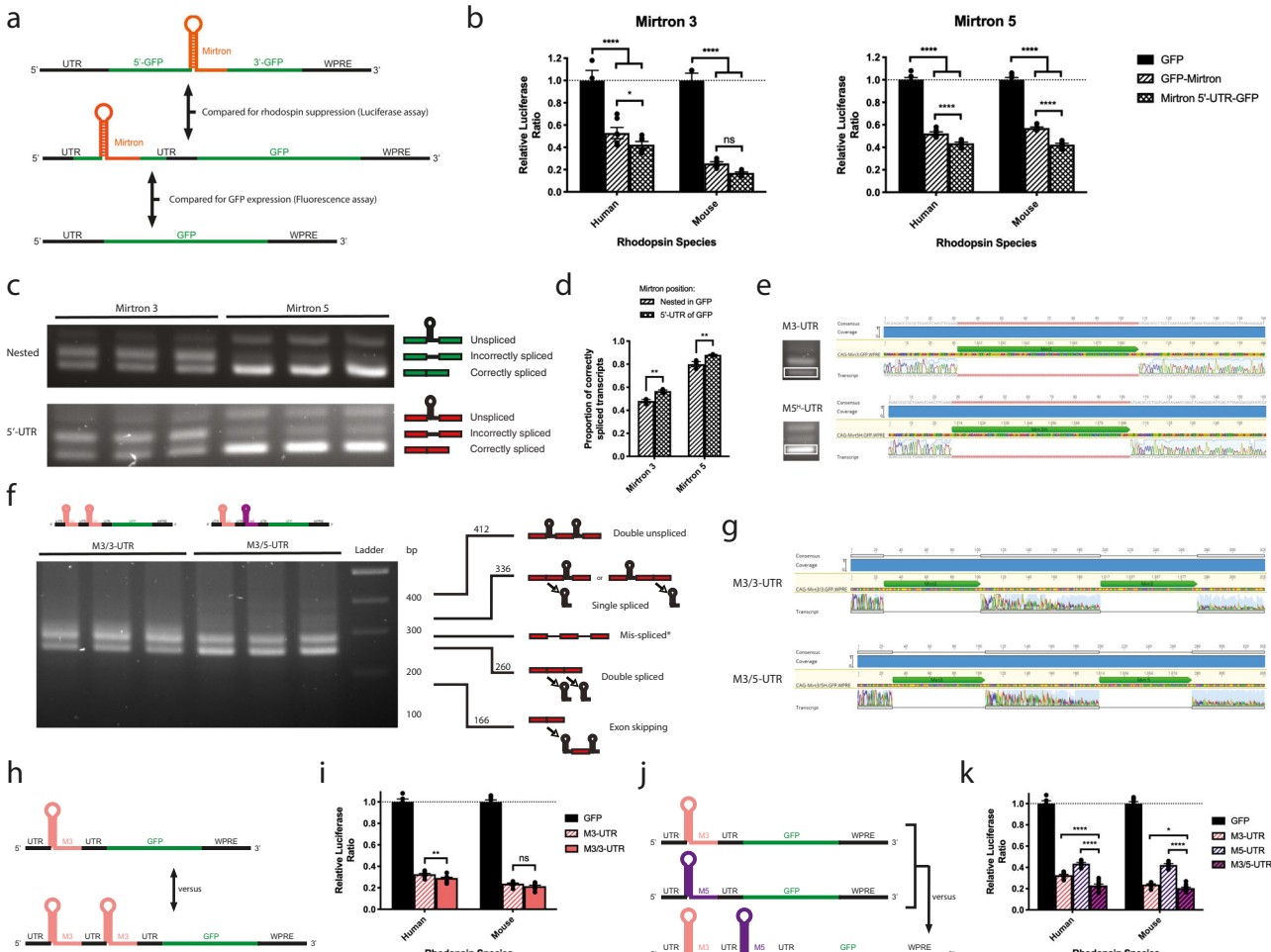

**Fig. 2 Splicing and potency of mirtrons located in the 5′-UTR of the *eGFP* transgene. a** Experimental scheme for comparison of nested and 5′-UTR mirtrons in HEK293 cells. **b** Rhodopsin knockdown compared for 5′-UTR mirtrons and nested equivalents. Data shown for M5 version directed against its corresponding rhodopsin species ($n = 6$). ns $p = 0.118$, *$p = 0.046$, ****$p < 0.0001$. **c** PCR products generated using mirtron-spanning primers and template cDNA derived from cells transfected with nested or 5′-UTR versions of mirtrons ($n = 3$). **d** Band densitometry revealed a greater proportion of correctly spliced transcripts for mirtrons located in the 5′-UTR ($n = 3$). Mirtron 3: $p = 0.0048$; Mirtron 5: $p = 0.0066$. **e** Sanger sequencing of the smallest bands (highlighted in white boxes) confirms accurate splicing of mirtrons from the 5′-UTR. Alignment of amplicon electropherograms with their corresponding reference sequences confirms absence of the 76 bp mirtron. **f** PCR splice analysis of tandem mirtrons ($n = 3$). Schematics illustrate predicted band sizes of splice products. *Band may represent a mixture of products. **g** Sequencing of the smallest bands in (**f**) confirms accurate splicing of tandem mirtrons. **h** Experimental scheme for comparison of M3-UTR and M3/3-UTR constructs. **i** Dual Glo® assay comparing M3-UTR and M3/3-UTR ($n = 12$). ns $p = 0.0583$, **$p = 0.0031$. **j** Experimental scheme for comparison of M3-UTR, M5-UTR and M3/5-UTR constructs. **k** Dual Glo® assay comparing M3-UTR ($n = 12$), M5-UTR ($n = 6$) and M3/3-UTR ($n = 12$). *$p = 0.0192$, ****$p < 0.0001$. Data for each version of M5-UTR and M3/5-UTR directed against its corresponding rhodopsin species are shown. M3/5^M-UTR did not provide additional knockdown of human *RHO* beyond that of M3-UTR. Similarly, no difference in mouse *Rho* suppression was detected between M3/5^H-UTR and M3-UTR. Mean ± SEM plotted throughout. Two-sided Šidák's multiple comparisons tests used in all cases.

(M3 and M5) were exchanged, where possible, for alternative sequences encoding the same amino acids (synonymous codons). Human and mouse codon frequencies were considered, and rare codons avoided[34]. The 21 bp M3 target site of human *RHO* was thus modified from CTTCCCCATCAACTTCCTCAC to ATTTCCAATTAATTTTCTGAC, whilst the human/mouse M5 target site was modified from AACGAGTCTTTTGTCATCTAC/ AACGAATCCTTTGTCATCTAC to AATGAATCCTTCGTGATTTAT. These changes conferred complete resistance of the resulting rhodopsin sequences to degradation by corresponding mirtrons (Fig. 3a). The M3/M5-resistant codon-modified human rhodopsin sequence (RHO^M3/5R) was able to drive protein expression in cell culture (Fig. 3b, c), and when packaged within an AAV under the control of a human rhodopsin promoter (the AAV-M3.M5^H.RHO^M3/5R vector) and delivered subretinally in

*Nrl.GFP/+, Rho^−/−* mice, a rescue of the rod-derived electro-retinogram (ERG) was achieved confirming functionality of the codon-modified transgenic protein (Fig. 3d). The ERG response of treated eyes was variable. Of the four treated eyes which gave the lowest signal, two had documented surgical complications: one had substantial transscleral reflux of the vector observed at the time of injection resulting in a shallow retinal detachment, whilst in another significant intraoperative subretinal haemorrhage was noted. All animals showing high ERG signal after treatment had documented uncomplicated good quality injections. In two of the treated eyes with no documented intraoperative complications, a low ERG signal was recorded. We attribute this to possible trans-scleral reflux of the vector after completion of the injection. Immunohistochemistry (IHC) revealed that treated retinas had OS that were absent in

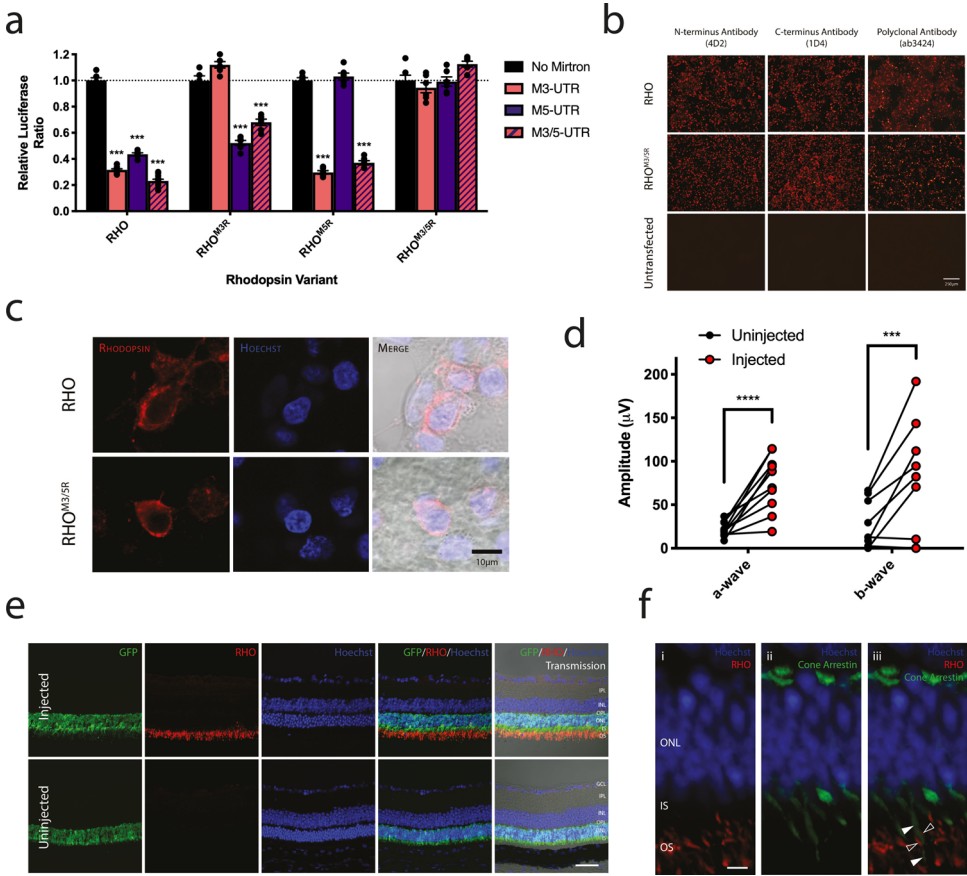

**Fig. 3 Codon modification of human rhodopsin. a** Dual Glo® assays demonstrate codon-modification of target regions confers resistance to suppression by the corresponding mirtron(s). Mean ± SEM plotted, n = 6. ****p < 0.0001, two-sided Dunnett's multiple comparisons test. RHO$^{M3R}$: M3-resistant *RHO*; RHO$^{M5R}$: M5-resistant *RHO*; RHO$^{M3/5R}$: M3- and M5-resistant *RHO*. **b** Rhodopsin immunostaining of wild type *RHO*- and codon modified *RHO$^{M3/5R}$*-transfected HEK293 cells (n = 3). Scale bar = 250 μm. **c** High magnification view confirming RHO and RHO$^{M3/5R}$ traffic in a similar fashion when expressed in HEK293 cells. **d** ERG confirms functionality of human transgenic rhodopsin protein expressed from the codon modified *RHO$^{M3/5R}$* transcript following subretinal delivery of AAV-M3.M5$^H$.RHO$^{M3/5R}$ in rhodopsin-null mice. Dark-adapted a- and b-wave responses of paired uninjected (left) and injected (right) eyes (n = 10). A-wave flash intensity: 10 cd.s/m$^2$ representing a mixed rod-cone response. b-wave responses followed dim flashes of 0.0001 cd.s/m$^2$ and were thus predominantly rod-derived. ***p = 0.0003, ****p < 0.0001, two-sided Šidák's multiple comparisons test. **e** Transgenic human rhodopsin is expressed in rods of *Nrl.GFP/+, Rho$^{-/-}$* mice following subretinal injection of AAV-M3.M5$^H$.RHO$^{M3/5R}$. Retinal cryosections from injected and uninjected eyes of an individual mouse stained for rhodopsin. n = 6. Scale bar = 200 μm. IPL: inner plexiform later, INL: inner nuclear layer, OPL: outer plexiform layer, ONL: outer nuclear layer, IS: inner segments, OS: outer segments. **f** High magnification section of an injected retina double stained for rhodopsin and cone arrestin. No overlap of staining was detected confirming restriction of transgenic protein expression to rods. The IS/OS of a single cone stained for cone arrestin (solid arrows) located between two rod OS expressing transgenic human rhodopsin (open arrows) is shown, suggesting rod-specific expression. Scale bar = 20 μm.

uninjected fellow eyes. These were packed with rhodopsin (Fig. 3e). No ectopic rhodopsin expression (i.e. outside of the rod OS) was identified (Fig. 3f). Thus, human rhodopsin protein translated under the control of a rod cell-specific promoter from an mRNA transcript resistant to both mirtrons 3 and 5 traffics to the OS of rods, and is capable of driving the dark-adapted light response.

**Splicing efficiency of artificial mirtrons in vivo**. To confirm that mirtrons are liberated from transgenic mRNA in vivo, *Nrl.GFP/+, Rho$^{P23H/+}$* mice were injected subretinally with an AAV vector (pseudotype AAV2/8 with Y733F capsid mutation[35]) combining M3 and M5$^H$ in tandem within the 5′-UTR of RHO$^{M3/5R}$ (AAV-M3.M5$^H$.RHO$^{M3/5R}$; Fig. 4a). The *Nrl.GFP/+, Rho$^{P23H/+}$* knock-in mouse carries the dominant P23H rhodopsin mutation along with a rod-specific eGFP reporter which permits en face imaging of the surviving rod cell population in vivo[36]. Right eyes were injected with vector at either low (2 × 10$^8$gc) or high dose

(2 × 10$^9$gc), whilst left eyes received subretinal sham injections of phosphate-buffered saline (PBS). Four weeks later, retinas were harvested and cDNA derived. Subsequent PCR analysis using mirtron-spanning primers revealed predominant bands corresponding in size to those expected from transcripts in which both mirtrons had been individually spliced (Fig. 4b, c). Correct splicing in vivo was confirmed by Sanger sequencing of M3 (Fig. 4d).

**Mirtron-derived miRNA expression and rhodopsin RNA replacement in vivo**. Small RNA expression analysis was performed using whole retinal samples 4 weeks after unilateral subretinal delivery of AAV-M3.M5$^H$.RHO$^{M3/5R}$ in *Nrl.GFP/+, Rho$^{P23H/+}$* mice (n = 5 for each low dose and high dose). Retinas from sham (PBS)-injected fellow eyes were processed in a similar fashion as internal controls. Sequence reads of 15–31 nucleotides were searched against micro RNAs in the microRNA database (miRbase 22), and all unannotated sequences representing potentially novel miRNAs were quantified and analysed. The 21

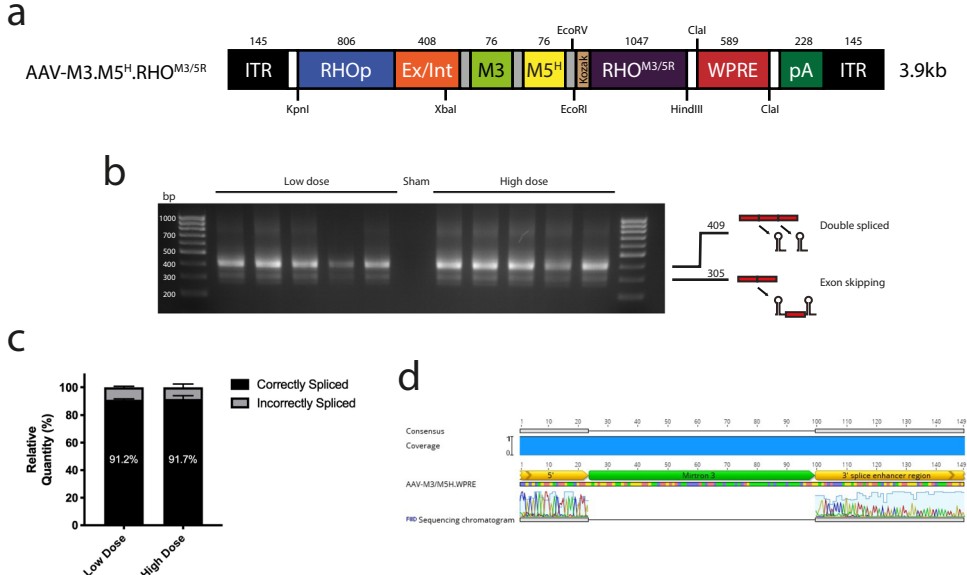

**Fig. 4 Artificial mirtrons delivered by a recombinant AAV vector are efficiently spliced from transgenic pre-mRNA in vivo. a** Genome map of AAV-M3. M5$^H$.RHO$^{M3/5R}$. Size in bp is indicated above each element and locations of relevant restriction sites marked. Size of the entire genome is shown to the right. ITR: inverted terminal repeat, RHOp: human rhodopsin promoter, Ex/Int: Exon/intron/exon splice site from the CAG promoter, RHO$^{M3/5R}$: codon-modified RHO coding sequence, WPRE: woodchuck hepatitis virus post-transcriptional regulatory element, pA: bovine poly-adenosine signal. **b** Artificial mirtrons delivered by a recombinant AAV vector are efficiently spliced from transgenic mRNA in vivo. DNA gel electrophoresis of amplicons generated using mirtron-spanning primers with template cDNA derived from AAV-M3.M5$^H$.RHO$^{M3/5R}$-injected *Nrl.GFP/+, Rho$^{P23H/+}$* retinas ($n = 5$). Schematics which illustrate the predicted origins of the bands detected, together with their size in bp, are shown to the right of the figure. The predominant band was of the size expected for transcripts from with which both mirtrons had been individually spliced. **c** Mean relative band density ± SEM for low ($2 \times 10^8$gc) and high dose ($2 \times 10^9$gc) samples ($n = 5$). Displayed figures relate to the percentages of amplicons predicted to have arisen from correctly spliced transcripts. **d** Sanger sequencing of DNA extracted from the predominant (larger) band confirmed accurate splicing of the 76 bp M3 sequence. Sequencing of the M5 region, and of the smaller of the two bands, was unreliable and data are not shown.

bp guide strand sequence for both M3 and M5$^H$ were identified in significant quantities in all AAV-M3.M5$^H$.RHO$^{M3/5R}$-injected samples but not in sham-injected fellow eyes (Fig. 5a). Sequence analysis confirmed accurate production of the 21 bp mature miRNA species from guide strands of both M3 (Fig. 5b) and M5$^H$ (Fig. 5c). In both cases, the 5′ end was more precisely generated (through splicing) than the 3′ end (generated by Dicer[25]). Short RNA corresponding to the passenger strand of M3 was detected in very low quantities by comparison (Fig. 5b), whilst none was detected for M5$^H$ (Fig. 5c). Levels of endogenous *Rho* knockdown and transgene-derived *RHO* replacement were determined by RT-qPCR using the same samples. Whole retinal gene expression studies revealed mean ± SEM knockdown of endogenous rhodopsin in eyes injected with $2 \times 10^9$ gc AAV-M3.M5$^H$.RHO$^{M3/5R}$ of 34.1 ± 5.0% compared with fellow sham-injected contralateral eyes (Fig. 5d; $p = 0.0024$). No rhodopsin knockdown was detected when an equivalent mirtronless vector (AAV-RHO; referred to as AAV-Ex/Int in Orlans et al. 2020[23]) was injected using the same experimental protocol (Fig. 5e). AAV-M3.M5$^H$.RHO$^{M3/5R}$ was capable of driving expression levels of human rhodopsin comparable to those of native mouse rhosopsin in sham-injected fellow eyes (Fig. 5f). For eyes injected with the AAV-M3.M5$^H$.RHO$^{M3/5R}$, but not for the mirtronless equivalent, a strong negative correlation existed between expression levels of mouse and human rhodopsin (Fig. 5g, h). Further, M3-derived miRNA expression strongly correlated with the observed degree of mouse rhodopsin suppression (Fig. 5i). AAV-M3.M5$^H$.RHO$^{M3/5R}$ thus functions as a rhodopsin mRNA replacement vector with a clear dose-response relationship. This represents the first demonstration to date of gene suppression by an artificial mirtron in vivo.

**Effect of mirtron-mediated RNA replacement therapy in vivo.**
AAV-M3.M5$^H$.RHO$^{M3/5R}$ was injected at one of two doses, $2 \times 10^8$gc (low dose, $n = 17$) or $2 \times 10^9$gc (high dose, $n = 17$), into the right eyes of *Nrl.GFP/+, Rho$^{P23H/+}$* mice, whilst left eyes remained uninjected, acting as internal controls. The effect on retinal structure and function was determined longitudinally in vivo by spectral domain optical coherence tomography (SD-OCT)/confocal scanning laser ophthalmoscopy (cSLO) and ERG respectively. The effect of sham injection had been determined using an identical protocol in a previous study in which a slight reduction in SD-OCT photoreceptor layer (PRL) thickness (defined as the vertical distance between the outer plexiform layer and RPE bands[37]) and ERG signal was observed in sham-injected relative to fellow uninjected eyes attributable to the surgical procedure[23]. Treatment here with low dose AAV-M3.M5$^H$.RHO$^{M3/5R}$ by contrast resulted in a significant slowing of retinal degeneration in the *Nrl.GFP/+, Rho$^{P23H/+}$* mouse as recorded by SD-OCT (Fig. 6a, b), cSLO fluorescence imaging (Fig. 6c) and ERG (Fig. 6d, e), an effect which was not observed when the equivalent but mirtronless AAV-RHO vector was injected at the same dose (Fig. 6f). The effect of treatment on PRL thickness as a function of retinal location is shown in Supplementary Fig. 3 whilst detailed ERG data are shown in Supplementary Figs. 4 and 5. At high dose, an effect size between that observed for low dose and sham-injected cohorts was evident (Supplementary Figs. 3–5) which may be attributable to a degree of rhodopsin overexpression (Fig. 5b).

**Discussion**
In this study we have developed a mutation-independent approach to the treatment of rhodopsin-related ADRP using artificial mirtrons. We have shown that the resulting vector can achieve RNA replacement and a modest slowing of retinal degeneration in a rapidly-

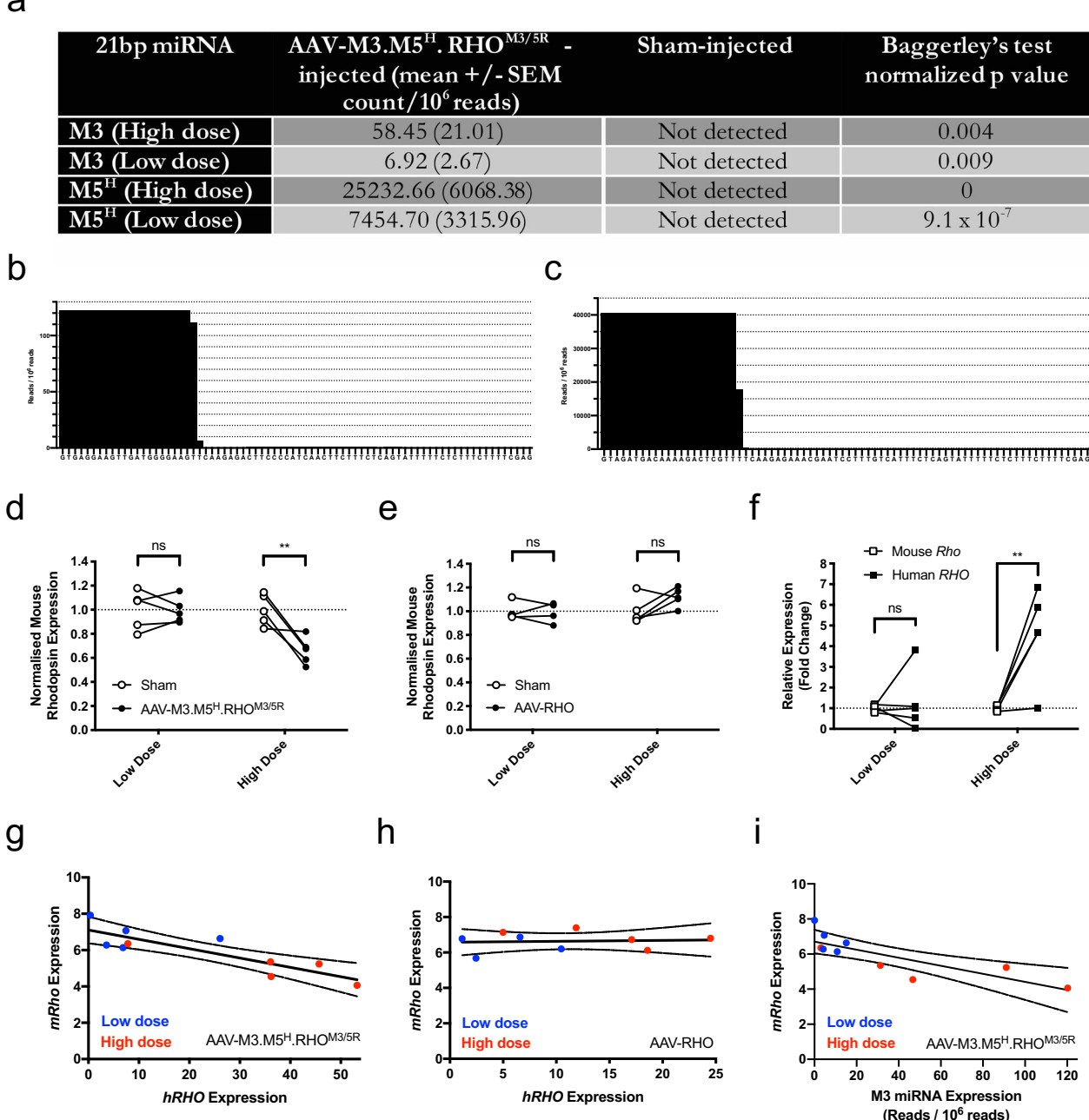

**Fig. 5 Micro RNA expression and gene expression analysis of knockdown and replacement arms of AAV-M3.M5$^H$.RHO$^{M3/5R}$ gene therapy 4 weeks after subretinal injection in the *Nrl.GFP/+, Rho$^{P23H}$/+* mouse.** Gene expression levels refer to the ratio *Rhodopsin: eGFP* (note eGFP expression is rod-specific in this model[36]). **a** Small RNA expression analysis detected mirtron-derived 21 bp mature miRNAs corresponding to Mirtron 3 (M3) and Mirtron 5$^H$ (M5$^H$) guide sequences in AAV-injected but not in sham-injected eyes. **b** Short RNA sequencing (15–31 nt) from high-dose AAV-M3.M5$^H$.RHO$^{M3/5R}$-treated retinas aligned to the Mirtron 3 sequence. **c** Short RNA sequencing (15–31 nt) from high-dose AAV-M3.M5$^H$.RHO$^{M3/5R}$-treated retinas aligned to the Mirtron 5$^H$ sequence. **d** Suppression of endogenous mouse rhodopsin gene expression by AAV-M3.M5$^H$.RHO$^{M3/5R}$. Paired right (AAV-injected) and left (sham PBS-injected) eyes are connected by joining lines. **$p = 0.0024$, two-way ANOVA, Šidák's multiple comparisons test. **e** No suppression of endogenous mouse rhodopsin was detected at either dose in eyes injected with the mirtronless AAV-RHO vector. Paired right (AAV-injected) and left (sham PBS-injected) eyes are connected by joining lines. Two-way ANOVA, Šidák's multiple comparisons test. **f** Human transgenic rhodopsin supplementation levels in AAV-M3.M5$^H$.RHO$^{M3/5R}$-injected retinas expressed as fold change relative to endogenous mouse rhodopsin in fellow eyes. **$p = 0.0011$, two-sided Šidák's multiple comparisons test. **g** Scatter plot of human *RHO* versus mouse *Rho* gene expression relative to *eGFP* housekeeper following subretinal injection of AAV-M3.M5$^H$.RHO$^{M3/5R}$ as determined by qPCR ($r^2 = 0.731$, $p = 0.0016$). Trend lines represent linear regression ± 95% confidence interval (dotted lines). **h** Scatter plot of human *RHO* versus mouse *Rho* gene expression relative to *eGFP* housekeeper following subretinal injection of AAV-RHO (mirtronless human rhodopsin vector) as determined by qPCR ($r^2 = 0.005923$, $p = 0.844$). **i** Scatter plot of mouse *Rho* gene expression as a function of Mirton-3-derived miRNA expression in eyes receiving subretinal injection of AAV-M3.M5$^H$.RHO$^{M3/5R}$ at low ($2 \times 10^8$gc) and high ($2 \times 10^9$gc) dose ($r^2 = 0.670$, $p = 0.004$).

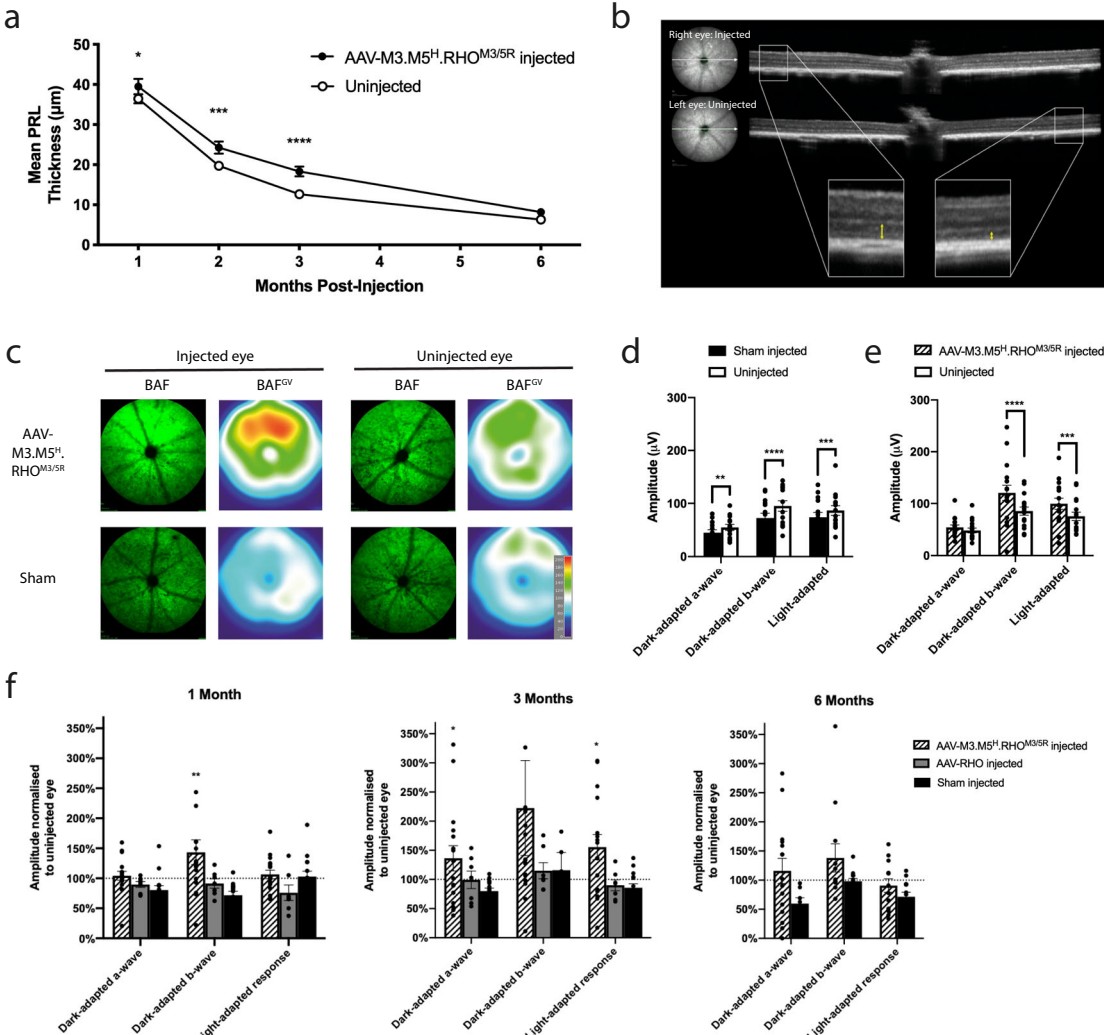

**Fig. 6 Partial rescue of retinal degeneration is observed following RNA replacement gene therapy in the rapidly degenerating *Nrl.GFP/+*, *Rho^{P23H}/+* mouse model of ADRP. a** Effect of subretinal injection of $2 \times 10^{8}$gc AAV-M3.M5H.RHO^{M3/5R} on PRL thickness (mean ± SEM; $n = 17$). Two way ANOVA for effect of injection $F(1, 15) = 6.999$; $p = 0.018$; *$p = 0.018$, ***$p = 0.0006$, ****$p < 0.0001$, Šidák's multiple comparisons test. **b** Representative SD-OCT images of right (treated) and left (untreated) eyes of a single animal taken one month following superior subretinal injection of $2 \times 10^{8}$gc AAV-M3.M5H. RHO^{M3/5R}. Orientation of sections is indicated by white arrows on corresponding en face infrared images shown to the left. Magnified views of equivalent points of temporal retina are shown in which the thickness of surviving PRL is highlighted by yellow arrows. **c** En face cSLO fluorescence imaging of *Nrl. GFP/+*, *Rho^{P23H}/+* retinas following subretinal injection of $2 \times 10^{8}$gc AAV-M3.M5H.RHO^{M3/5R} or PBS (sham). Representative blue autofluorescence (BAF) images and BAF mean grey value (BAF^{mGV}) surface plots taken 4 weeks post-treatment. Red signal represents increased density of surviving rods[36]. Paired right (injected) and left (uninjected) eyes are shown in each case. **d** Dark-adapted a- and b-wave and light-adapted ERG responses recorded 3 months after uniocular sham (PBS) injection (mean ± SEM; $n = 14$). a-wave measurements were recorded with flash intensity 10 cd.s/m² whilst b-wave responses followed dim flashes of 0.0001 cd.s/m². **$p = 0.0039$, ***$p = 0.0005$, ****$p < 0.001$, Šidák's multiple comparisons test. **e** Equivalent ERG responses recorded three months after uniocular treatment with $2 \times 10^{8}$gc AAV-M3.M5H.RHO^{M3/5R} (mean ± SEM; $n = 17$). **f** ERG signal of treated eyes normalised to that of fellow untreated eyes at 1-, 3- and 6-months post-injection (mean ± SEM; $n = 17$, 7 and 14 for AAV-M3.M5H.RHO^{M3/5R}, AAV-RHO and Sham groups respectively). One month **$p = 0.0073$; 3 months dark-adapted a-wave *$p = 0.0414$, light-adapted response *$p = 0.012$, Šidák's multiple comparisons test.

degenerating clinically relevant knock-in mouse model. The observed efficacy here may be directly attributable to the inclusion of rhodopsin-targeting mirtrons as an equivalent but mirtronless vector failed to impart any measurable benefit in a parallel study using the same experimental protocol[23]. Knockdown/replacement gene therapy has been demonstrated in transgenic models of ADRP using rhodopsin-targeting shRNAs under the control of Pol-III promoters together with codon-modified rhodopsin supplementation under rod-specific Pol-II promoter control. These studies have either employed separate suppression and supplementation vectors[14], or have combined the two therapeutic arms driven by separate

promoters within a single AAV[15,16,38]. Since various alternative mouse models of ADRP were used by these researchers, the magnitude of reported treatment effects cannot be compared directly with that reported here. In a recent study, Tsai et al. developed a novel dual vector CRISPR-Cas9-based knockdown/replacement treatment strategy, which they validated in the P23H and D190N knock-in mouse models of ADRP[39]. In spite of differences in the timing of treatment (these authors injected mice at post-natal day 1–3), the magnitude of functional rescue they observed in the Rho^{P23H}/+ model, as measured by ERG, is broadly comparable to that achieved with our mirtron-based vector.

Although introducing additional target sequence constraints, the application of artificial mirtrons for knockdown/replacement gene therapy confers several potential advantages over other RNAi-based approaches, namely the by passing of a potential molecular bottleneck (Drosha/DGCR8)[25], more accurate generation of the critical 5′ seed region of the guide strand (Fig. 5b, c)[24,29,40], and cell-specific RNAi expression[29,31]. In addition, the single-promoter construct developed here allows for consistent matching between the therapeutic arms within any given cell, which may reduce the risk of toxicity related to any imbalance between rhodopsin suppression and supplementation[21,22,41,42]. Direct comparison studies using equivalent vectors would be required to determine whether these theoretical advantages of mirtron-based knockdown/replacement gene therapy are borne out in practice.

Splicing efficiency for mirtrons M1–7 varied dramatically, from <1% (M1) to near 100% (M5). These designs, with the exception of M4, fulfilled the criterion of Seow et al. for design of artificial mirtrons that sequences commence with a GURRR motif[29]. M4 had a C at position +5 but was nevertheless spliced efficiently from its parent transcript. Conversely, M1 met this criterion but very little splicing was detected from the intended 5′ donor site. The presence of three purines at positions +3 to +5 did not therefore predict robust splicing of mirtrons in this study. Further, the 5′ donor splice sites of the efficiently spliced mirtrons M5 and M7 differed between positions +4 and +6 from those that define the canonical splice consensus of GURAGU[43] by 2 and 1 nucleotides respectively. This region of the intron is critical for donor splice site recognition by small nuclear RNA components of the spliceosome[44,45]. Taken together, these data would suggest that a variety of splice donor sites can result in effective liberation of mirtrons from transcripts, and that splicing efficiency of artificial mirtrons must be determined empirically.

A proportion of transcripts for all seven mirtron designs spliced incorrectly as a result of cryptic splice donor/acceptor sites. The presence of alternative splice products provides a compelling argument for locating mirtrons within the 5′-UTR rather than the CDS of a transgene. In the latter case, mis-splicing could result in production of mutant protein or NMD for frame shifts. Mis-splicing of mirtrons within the 5′-UTR would however have minimal impact on downstream gene expression, and importantly would not affect transgenic protein structure.

In their in vitro study, Curtis et al. combined artificial mirtrons with a replacement transgene under the control of a single Pol-II promoter as part of a knockdown/replacement gene therapy construct for spinocerebellar ataxia 7[31]. The authors found that mirtrons that spliced efficiently from within the CDS of GFP failed to do so in other contexts, stipulating that the mirtron's hairpin secondary structure may have restricted spliceosome access. They further postulated that a lack of exonic splice enhancers (ESEs: hexameric motifs within exons that support accurate and efficient splicing of nearby introns) at alternative locations may have contributed to this effect. Here, 5′ and 3′ flanking regions of the eGFP reporter in which splicing and potency of mirtrons had initially been validated were included alongside the mirtron when cloned into the 5′-UTR. This ensured that the ESE-rich context from which mirtrons were so effectively spliced was preserved. This strategy is only possible for mirtron placement in the UTR region of a transcript as additional nucleotides flanking a mirtron located within the CDS would disrupt translation post-splicing. Placement within the 5′-UTR also allowed multiple mirtrons to be co-expressed alongside the replacement gene. Curtis et al. found that delivery of multiple mirtrons as separate introns within the CDS of a transgene adversely affected splicing efficiency[31]. This may have explained why they observed no significant increase in overall potency for double- versus single-mirtron constructs. In this study, 5′-UTR mirtrons were individually and accurately spliced and had additive suppressive effects against rhodopsin targets. Inclusion of two or more independently functioning mirtrons in tandem within the 5′-UTR may be a useful general strategy for knockdown/replacement gene therapy application, as it potentially allows a construct to retain efficacy even when the point mutation carried by an individual falls within the one of the target regions. Tandem mirtrons could also be applied for the treatment of complex non-Mendelian disease, where they might be used to suppress the expression of multiple genes within a molecular pathway[28].

In this study a modest recue effect was observed following treatment of rapidly-degenerating adult mice with established disease. This contrasts with earlier studies in which younger mice with a slower degenerating phenotype were treated during the period of retinal development[14,38]. Treatment with $2 \times 10^8$ gc was found to be more effective than treatment with $2 \times 10^9$ gc of AAV-M3.M5$^H$.RHO$^{M3/5R}$ which may be due to a degree of retinal toxicity as a result of over-replacement of rhodopsin[23]. Optimisation of the AAV construct, for example through inclusion of additional rhosopsin-targeting mirtrons and/or weakening of the expression cassette to reduce human rhodopsin expression, may further improve efficacy of future iterations of the knockdown/replacement vector developed here. The human RHO-targeting version of M5 (M5$^H$) was chosen for use in the in vivo part of this study in anticipation of future human translation. This mirtron would not be expected to be active in mice (see Fig. 1f). The rhodopsin-suppressing effect of the vector described might therefore be expected to be greater when applied to human subjects. Further, since only rhodopsin target sites within areas of homology between human and mouse transcripts were considered in this study, it is possible that more effective species-specific mirtrons may be identified whose target sites fall outside of these regions.

In summary, this study represents the first in vivo demonstration of RNA replacement gene therapy using artificial mirtrons. This strategy has several advantages over existing techniques and may be broadly applied for the treatment of any dominantly-inherited disease.

## Methods

**Design of candidate mirtrons**. The core structure of mirtrons used in this study is based on the 3′-tailed design described and validated by Kock et al. 2015[30]. The advantage of artificial 3′-tailed over canonical mirtrons is that they have fewer sequence constraints by virtue of the intronic branch point and polypyrimidine tract being located outside of the stem-loop structure which contains the guide strand (Supplementary Fig. 6). For all artificial mirtrons used in this study, the guide strand was located in the 5′ arm of the hairpin stem, rather than the 3′ arm where cellular processing may be less accurate[31]. The guide strand was connected to a complementary 3′ passenger arm by a TTCAAGAGA loop motif. Mismatches between guide and passenger strands of the stem were purposefully introduced at the 5′ end to reduce local internal stability and facilitate correct selection of the guide strand (as opposed to the passenger strand) by the RNAi-induced silencing complex (Supplementary Fig. 7a–h)[46,47].

Potential targets for artificial mirtrons within the rhodopsin cDNA sequence were identified according to several criteria. Firstly, only areas of at least 21 bp in length with perfect or near-perfect homology between the mouse and human rhodopsin mRNA transcript were considered. This is so that mirtrons designed for incorporation into a treatment for ADRP in man could first be validated in a relevant mouse model. Secondly, since a 21 bp guide sequence located within the 5′ arm of a mirtron must commence with a splice donor motif, only potential target sequences that featured the reverse complement of such a motif at their 3′ end were considered. Thirdly, mirtron candidates that contained an additional strong splice donor motif elsewhere within the 21 bp guide sequence were not considered as such mirtrons would likely be preferentially spliced incorrectly.

Candidate mirtron sequences nested within the eGFP CDS were input into online splice prediction software (Spliceport: http://spliceport.cbcb.umd.edu and Human Splice Finder: http://www.umd.be/HSF3/HSF.shtml) to identify those that included strong donor motifs. Scores for the mirtrons that were subsequently manufactured and tested in vitro are shown in Supplementary Table 1. The siRNA at Whitehead online RNAi prediction tool (http://sirna.wi.mit.edu) was also used to identify which candidate sequences were likely to make effective RNAi targets.

With the above in mind, seven 3′-tailed mirtrons (designated M1–M7) were designed targeting different parts of the rhodopsin CDS (Supplementary Fig. 7). The 21 bp guide sequences of M1-4 and M6 were perfect antisense matches to both

the human and mouse transcripts. That of M7 was a complete antisense match for the human CDS, but contained a single mismatch with the mouse sequence. This mismatch would however be predicted to form a non-Watson & Crick G-U 'wobble' base pair at the RNA level, and so may nonetheless allow effective targeting by this mirtron[48]. The sequence targeted by M5 was identified by the siRNA at Whitehead online tool as a strong candidate for effective RNAi-induced cleavage. The mouse and human M5 targets however differed by two base pairs. Two versions of M5 were therefore created: M5[H] matching the reverse complement of the human CDS, and M5[M] matching that of the mouse sequence.

The guide strand and target sequences together with their respective target positions within the rhodopsin CDS are shown in Supplementary Table 2.

**Cloning**. Double stranded mirtron inserts with 5′ and 3′ overhangs complementary to those of the BstBI restriction site were engineered by annealing complementary oligonucleotides. Forward oligonucleotides comprised the desired 76 bp mirtron sequence with a 5′-P-CGAAG motif (where P represents a phosphate group) and corresponding omission of CGAAG from the 3′ end. Reverse oligonucleotides comprised the reverse complement of the desired mirtron sequence with omission of the 5′-CTT motif, a 5′ phosphate and inclusion of CTT at the 3′ end. Oligonucleotides were manufactured by Sigma-Aldrich, UK, and reconstituted to a concentration of 10 µM upon receipt. Ten µl forward and 10 µl reverse solution were thoroughly mixed and heated to a temperature of 95 °C for 5 min using a PCR thermocycler (Multigene, Labnet International Inc.). Samples were then slowly cooled in a stepwise manner to room temperature over a period of 6 min to facilitate efficient annealing. The resulting double stranded mirtron sequence (which by design already had BstBI adhesive ends) was ligated directly into the BstBI/shrimp alkaline phosphatase (both New England Biolabs, UK) treated CAG.eGFP.WPRE plasmid backbone. The *eGFP* CDS is shown below. The BstBI restriction site is underlined, with the cut point marked by a vertical line:

```
ATGAGCAAGGGCGAGGAACTGTTCACTGGCGTGGTCCCAATTCT
CGTGGAACTGGATGGCGATGTGAATGGGCACAAATTTTCTGTCA
GCGGAGAGGGTGAAGGTGATGCCACATACGGAAAGCTCACCCT
GAAATTCATCTGCACCACTGGAAAGCTCCCTGTGCCATGGCCAA
CACTGGTCACTCACCTGACCTATGGCGTGCAGTGCTTTTCCAGA
TACCCAGACCATATGAAGCAGCATGACTTTTTCAAGAGCGCCAT
GCCCGAGGGCTATGTGCAGGAGAGAACCATCTTTTTCAAAGAT
GACGGGAACTACAAGACCCGCGCTGAAGTCAAGTT|CGAAGGTG
ACACCCTGGTGAATAGAATCGAGCTGAAGGGCATTGACTTTAAG
GAGGATGGAAACATTCTCGGCCACAAGCTGGAATACAACTATAA
CTCCCACAATGTGTACATCATGGCCGACAAGCAAAAGAATGGCA
TCAAGGTCAACTTCAAGATCAGACACAACATTGAGGATGGATCC
GTGCAGCTGGCCGACCATTATCAACAGAACACTCCAATCGGCGA
CGGCCCTGTGCTCCTCCCAGACAACCATTACCTGTCCACCCAGT
CTGCCCTGTCTAAAGATCCCAACGAAAAGAGAGACCACATGGTC
CTGCTGGAGTTTGTGACCGCTGCTGGGATCACACATGGCATGGA
CGAGCTGTACAAGTGA
```

Correct orientation of the mirtron within the construct was confirmed by commercial Sanger sequencing with primer sequence 5′-CACAAATTTTCTGTCAGC (Source Biosciences, UK).

To evaluate the efficacy of mirtrons, human and mouse rhodopsin target sequences were subcloned from plasmid DNA and murine retinal cDNA respectively into the multiple cloning site (MCS) of the PsiCHECK2 vector (Promega Inc., UK) in accordance with manufacturer's instructions. This plasmid is designed to provide a rapid approach for the evaluation of RNAi in vitro. The plasmid contains two luciferase genes- *Renilla* and firefly- both under the control of strong ubiquitous promoters. The *Renilla* luciferase has an MCS located between its translational stop codon and polyA sequence into which the target gene of interest is cloned. Cleavage of the resulting mRNA transcript by an RNAi effector will result in its deadenylation and degradation. This would be expected to result in a reduction of the recorded *Renilla* luciferase signal. The firefly luciferase signal acts as an internal control to which the *Renilla* luciferase signal is normalised.

To generate the PsiCHECK2-RHO vector, a human rhodopsin amplicon was amplified by PCR using CAG.RHO.WPRE plasmid template and forward and reverse primers with sequences 5′-ATCTCGAGATGAATGGCACAGA and 5′-ATGCGGCCGCTTAGGCCGGGGCCA respectively. The resulting 1065 bp DNA fragment, identified by agarose gel electrophoresis, was extracted and digested with XhoI and NotI enzymes (both New England Biolabs, UK). The digestion product was then ligated into the XhoI/NotI-digested PsiCHECK2 vector and the correct sequence of the resulting PsiCHECK2-RHO plasmid confirmed by commercial Sanger sequencing. To generate the PsiCHECK2-mRho vector, a mouse rhodopsin amplicon was amplified by PCR using mouse retinal cDNA template and forward and reverse primers 5′-ATGCGATCGCATGAACGGCACAGA and 5′-ATGCGGCCGCTTAGGCTGGAGCCA respectively. The resulting 1065 bp DNA fragment identified by agarose gel electrophoresis was extracted and digested with AsiSI and NotI enzymes (both New England Biolabs, UK). The digestion product was then ligated into the AsiSI/NotI-digested PsiCHECK2 vector and the correct sequence of the resulting PsiCHECK2-mRho plasmid confirmed by commercial Sanger sequencing using a primer with sequence 5′- TCGAGTCCGACCCTGGGTTCT. A complete list of primers used in this study is shown in Supplementary Table 3.

**Optimisation of 5-UTR mirtron flanks**. Splicing of mirtrons is context dependent and reliant of the presence of nearby exonic splice enhancer sequences (ESEs). To encourage efficient splicing of mirtrons within the 5′UTR, flanking sequences around the original mirtron insertion site within *eGFP* were also included (i.e. around the BstBI restriction site where efficient splicing had already been confirmed), as these were found to contain numerous ESEs (as identified using the online RESCUE-ESE tool[49], http://genes.mit.edu/burgelab/rescue-ese/). An in silico splice analysis was performed using online software (Spliceport, http://spliceport.cbcb.umd.edu) to identify the optimal 5′ and 3′ flank length, whilst avoiding 'ATG' motifs which may act as cryptic translational start sites[50]. Supplementary Table 4 shows the splice scores calculated using the Spliceport online tool for Mirtron 3 with flanks from the neighbouring *eGFP* CDS of different sizes when placed at the intended insertion site within the 5′-UTR: an EcoRV restriction site (which creates a blunt-ended cut) between the CAG promoter and Kozak consensus sequence.

The maximum 5′ flank size which avoided an 'ATG' motif was 41 bp and this gave the highest donor consensus score. The 3′ flank length had little effect on acceptor score which was found to be strong in all cases. Upstream flank size did not have a significant effect on acceptor score, and downstream flank size similarly had little effect on donor score. Thus, a 41 bp upstream flank and 53 bp downstream flank from the *eGFP* CDS around the BstBI cut site was used and amplified using primers 5′-TGACGGGAACTACAAGACCC and 5′- ATCCTCCTTAAAGTCAATGCCC. The sequences of the upstream (5′) and downstream (3′) flanks are shown in Supplementary Fig. 8 in black, whilst the positions of ESE hexamer motifs identified using the RESCUE-ESE online tool are shown above each in red.

**Cell culture and in vitro transfection**. Human Embryonic Kidney 293 (HEK293) cells (#85120602, Culture Collections, Public Health England) were cultured in complete Dulbecco's Modified Eagle Medium (complete DMEM) consisting of DMEM supplemented with 10% filtered foetal bovine serum, L-glutamine (2 nM), penicillin (100 U/ml), streptomycin (100 µg/ml), 1% non-essential amino acids, and phenol red pH indicator (Gibco, Thermo Fisher Scientific, UK). Prior to transfection, cells were seeded at a density of $10^5$ cells/ml and incubated in multiwell plates overnight. Transfection was performed using *TransIT*® LT-1 reagent (Mirus Bio, USA) in accordance with manufacturer's instructions.

**Fluorescence spectroscopy**. Fluorescence spectroscopy assays were performed for quantification of splicing for mirtron constructs nested within the *eGFP* CDS, and for assessment of the effect of 5′-UTR mirtrons on downstream transgene expression. For the splicing assay, HEK293 cells were seeded into 12 well plates and transfected with 1 µg pCAG.eGFP.WPRE or pCAG.eGFP-M1-7.WPRE. For gene expression assays, cells seeded in 12 well plates were transfected with $7.22 \times 10^{10}$ copies of pCAG.eGFP.WPRE or Mirtron-UTR plasmid. In either case, wells were photographed 48 h later using standardised acquisition settings, after which cells were pelleted and resuspended in 100 µl of fluorescence immunoprecipitation (FIPA) buffer (NaCl 150 mM, Tris base 50 mM, EDTA 2 mM, 1% TritonX-100; adjusted to pH7.4 with HCl) supplemented with protease inhibitor (PI, Roche Diagnostics, one tablet per 10 mls). Total protein was quantified from lysates using the Pierce BCA Protein Assay Kit (Thermo Fisher Scientific, UK) in accordance with manufacturer's instructions. This colourimetric assay quantifies protein of appropriately diluted samples against a standard curve of bovine serum albumin of known concentration. All reactions were performed in duplicate, and colourimetric absorbance readings taken at 562 nm using the iMark™ microplate reader (Bio-Rad, UK). For each sample, 10 µg of protein was loaded into wells of a black plastic 96 well plate (Greiner Bio-One, UK) in duplicate and the volume made up to 300 µl with FIPA buffer/PI. Fluorescence (in arbitrary units) was measured using the Viktor[3] 1420 Multilabel plate reader (Perkin Elmer, UK), with excitation at 485 nm, and detection at 535 nm and an exposure time of 1 s per well. To account for background fluorescence, the mean reading taken from protein lysates derived from untransduced cells was subtracted from all transduced cell fluorescence readings. Signals obtained from protein lysates extracted from pCAG.eGFP-M1-7.WPRE, or Mirtron-UTR transfected cells were then normalised to that recorded from pCAG.eGFP.WPRE transfected cell lysates to obtain a measure of splicing efficiency, or normalised reporter gene expression respectively.

**Luciferase assays**. For the Dual Glo® assay, HEK293 cells were seeded into 96 well culture plates. After 24 h, cells were co-transfected with 65 ng mirtron plasmid and 32.5 ng PsiCHECK2-target plasmid. Forty-eight hours later, 30 µl of media was removed from each well and the assay performed in accordance with manufacturer's instructions. Briefly, 70 µl of *Renilla* luciferase substrate was added to each well and the plate left to incubate for 20 min. *Renilla* luminescence signal was then recorded (in arbitrary units) using the Viktor[3] plate reader. Seventy µl of Stop & Glo reagent (which quenches *Renilla* luciferase activity and includes the substrate for Firefly luciferase) was then added to each well and the plate again left to incubate at room temperature for 20 min. Firefly luminescence was then similarly read (in arbitrary units) using the Viktor[3] plate reader. Mirtron-mediated mRNA knockdown was estimated by calculating the relative luciferase ratio *Renilla*/Firefly, which was then normalised to that recorded from cell samples transfected with the mirtronless CAG.GFP.WPRE plasmid.

**Immunocytochemistry**. Immunocytochemistry was performed on HEK293 cells cultured in chamber slides (Nunc Lab-Tek II chamber slide system™, Thermo Fisher

Scientific, UK) 48 h after transfection. Media was removed from the cell monolayer and two PBS washes were applied, each consisting of the gentle addition and removal of 120 µl of 0.01 M PBS. Cells were fixed in 120 µl 4% formaldehyde (Thermo Fisher Scientific, UK) for 15 min at room temperature followed by two further PBS washes as above. Cells were then permeabilised by incubation in 120 µl 0.2% Triton X-100 in 0.01 M PBS for ten minutes, washed once in PBS and incubated in 120 µl 10% normal donkey serum (NDS) with 0.1% Triton X-100 in 0.01 M PBS for 30 min. After a further PBS wash, the cell monolayers were incubated in 100 µl primary antibody diluted 1:1000 in 1% NDS with 0.1% Triton X-100 in 0.01 M PBS for 2 h at room temperature. Four PBS washes were then applied and the cell monolayers were incubated in 100 µl secondary antibody (conjugated with fluorescent dye) diluted 1:200 in 1% NDS with 0.1% Triton X-100 in 0.01 M PBS for 30 min at room temperature in the dark. Three PBS washes were applied after which cells were counterstained through incubation in 120 µl Hoechst 33342 (Life Technologies, USA) diluted 1:1000 in 0.01 M PBS for 10 min. Slides were mounted using Prolong Diamond anti-fade medium (Thermo Fisher Scientific, Hemel Hempstead, UK). Details of antibodies used are outlined in Supplementary Table 5.

**Animal models.** Mice were housed in a dedicated facility at 20–24 °C and 45–65% humidity with a 12 h light/dark cycle and food and water available ad libitum. All animal procedures were conducted in accordance with the Animals (Scientific Procedures) Act 1986, UK and with the Association for Research in Vision & Ophthalmology (ARVO) statements on the care and use of animals in ophthalmic research under a UK Home Office Personal and Project License. All such experiments were approved by the University of Oxford Animal Ethics Committee. Two transgenic species were used in this study. The first, designated *Nrl.GFP/+, Rho*$^{-/-}$, was a rhodopsin knockout mouse with a GFP transgene under the control of the *Nrl* promoter. The second, designated *Nrl.GFP/+, Rho*$^{P23H/+}$, was a knock-in mouse carrying a heterozygous P23H mutation at the rhodopsin locus, also with an a GFP transgene under the control of the *Nrl* promoter. Inclusion of the *Nrl.GFP* transgene in these models allowed dynamic en face imaging of the surviving population of rods using blue autofluorescence imaging cSLO. Detailed descriptions of these models and techniques may be found elsewhere[36].

**In vivo procedures.** For all in vivo procedures, mice were anaesthetised with a combination of ketamine (Vetalar, Boehringer Ingelheim, Bracknell, UK; 80 µg/g body weight) and xylazine (Rompun; Bayer, Reading, UK; 10 µg/g body weight) diluted in sterile 0.9% saline, delivered by intraperitoneal injection. Pupils were then dilated with tropicamide 1% and phenylephrine 2.5% eye drops (both Bausch & Lomb, Kingston upon Thames, UK). Where appropriate anaesthesia was reversed at the end of procedures by intraperitoneal injection of atipamezole (Antisedan, Zoetis, Leatherhead, UK; 2 mg/kg body weight) diluted in sterile 0.9% saline.

**Subretinal injections.** Animals in this study received single unilateral subretinal injections of PBS ('Sham') or AAV at a dose of either $2 \times 10^8$ gc/ml ('Low dose') or $2 \times 10^9$ gc/ml ('High dose'), with total volume of 1.5 µl in all cases. All injections were delivered in the superior sector by a transscleral approach in mice aged postnatal week 3[36]. Prior to injection, animals were anaesthetised and pupils dilated as described above. A drop of proxymetacaine (Minims, Bausch & Lomb, Kingston upon Thames, UK) was applied and a corneal paracentesis was performed by brief puncture using a 33G needle to allow efflux of a small bleb of aqueous. Carbomer gel (Viscotears Liquid Gel, Alcon, Camberley, UK) was placed on to the corneal surface followed by a 5 mm glass coverslip which allowed direct visualisation of the posterior pole through the operating microscope. The transscleral subretinal injection was then performed by passing a 35G beveled NanoFil needle mounted on a Nanofil 10 µl syringe (both World Precision Instruments, Hitchin, UK) into the subretinal space under direct visualisation. Following the procedure, the glass coverslip was removed and a drop of chloramphenicol 0.5% (Bausch & Lomb, Kingston upon Thames, UK) was applied to the corneal surface.

**Ocular imaging.** Widefield retinal imaging was performed with a 55° lens on the Spectralis imaging platform (Heidelberg Engineering, Heidelberg, Germany). Following anesthesia and pupillary dilatation, Hypromellose 0.3% (Blumont Healthcare Ltd., United Kingdom) drops were applied and polymethyl methacrylate contact lenses (Cantor & Nissel Ltd., Brackley, United Kingdom) placed on the corneal surface bilaterally. Near-infrared (815 nm diode laser) and blue autofluorescence (BAF; 486 nm blue diode excitation laser with a 500 nm barrier filter) retinal images were acquired using the confocal scanning laser ophthalmoscope (cSLO) focused in the plane of the RPE and centered on the optic nerve head. BAF images were acquired at a sensitivity value of 50 without image normalisation. All fundus images were high resolution and acquired in automatic real-time mode. Eight equally spaced spectral domain ocular coherence tomography B-scans were acquired in a radial orientation centered on the optic nerve head for each eye. Final images were produced from an average of 25 individual B-scans. Photoreceptor layer (PRL) thickness was defined as the vertical distance between the outer plexiform layer and RPE bands. PRL thickness was measured using in-built calipers at eight locations (superior, superonasal, nasal, inferonasal, inferior, inferotemporal, temporal and superotemporal) at a distance of

22.5° from the optic nerve margin. Mean PRL thickness for each eye was calculated from all eight measurements.

**Electroretinography.** Animals were dark-adapted for a minimum of 12 h prior to electroretinography (ERG) within a purpose-built light-tight cabinet. Following anesthesia and dilation of pupils, mice were positioned on a heated platform and subcutaneous ground and reference electrodes placed on the flank and between the eyes respectively. Achromatic ACLAR custom contact lenses (Honeywell International, Charlotte, NC, USA) were used to hold silver Dawson, Trick and Litzkow Plus electrodes (Diagnosys LLC, Cambridge, UK) on each cornea. The platform holing the mouse was then positioned to the centre of a Ganzfeld stimulator (Colordome; Diagnosys LLC, Cambridge, UK). The protocol for ERG consisted of dark- and light-adapted phases and was run using Espion E3 software (Diagnosys LLC). During the dark-adapted protocol, ERG responses were provoked by brief uniform flashes of white light at a range of intensities between $-6$ and 1.4 log cd.s/m$^2$, increasing in log unit steps. At each level of luminance, up to 16 measurements were acquired that were later averaged prior to analysis. Following completion of the dark-adapted phase, animals were exposed to a full-field white light of 30 cd/m$^2$ for 10 min to induce light-adaption. The subsequent light-adapted protocol consisted of superimposed bright white flashes of intensity between 0.3 and 10 cd.s/m$^2$ increasing in half log unit steps. A total of 25 recordings were taken at each step and averaged.

**Immunohistochemistry.** Follow euthanasia, eyes for IHC were extracted and the anterior segments (cornea, iris, and lens) were removed. Eye cups were fixed in 4% formaldehyde for 30 min and then moved sequentially into 10%, 20%, and 30% sucrose solutions, for at least 20 min each. The eye cups remained in 30% sucrose solution overnight at 4 °C, were washed in PBS, dried, and then submerged in optimal cutting temperature compound (VWR International, Lutterworth, United Kingdom) within a mold. The molds were transferred to dry ice to induce freezing and the specimens cut to a thickness of 18 microns using a cryostat. Sections were placed onto polylysine-coated glass slides and left to dry at room temperature overnight. The slides were subsequently incubated in 0.2% Triton X-100 in PBS for 20 min and then blocked in 10% NDS-PBS for a period of 1 h. Primary rabbit anti-rhodopsin antibody (ab3424; Abcam, Cambridge, United Kingdom) diluted 1:1,000 in PBS with 1% NDS was placed onto the sections, and slides were left to incubate overnight at 4 °C. Slides were then incubated with secondary fluorescence-tagged donkey anti-rabbit IgG-568 antibody (Alexa-Fluor ab175470; Abcam) diluted 1:500 in PBS for 2 h at room temperature. The sections were counterstained with Hoechst 33342 (Thermo Fisher Scientific, Hemel Hempstead, United Kingdom) and mounted with ProLong Diamond Antifade Mounting Medium (Thermo Fisher Scientific). Confocal imaging was performed using an LSM-710 inverted confocal microscope (Carl Zeiss, Cambridge, United Kingdom).

**Gene expression analysis.** RNA was extracted from neural retina using the miRNeasy Mini kit (Qiagen, Manchester, UK) in accordance with manufacturer's instructions. Genomic DNA contaminants were removed as part of this process using an RNase-free DNase set (Qiagen, Manchester, UK). RNA was reverse transcribed using the Invitrogen SuperScript III kit (Thermo Fisher Scientific, Hemel Hempstead, UK) and a starting quantity of 1 µg nucleic acid. TaqMan assays for human rhodopsin, mouse rhodopsin and EGFP (Thermo Fisher Scientific, Hemel Hempstead, UK) were applied using the resulting cDNA for evaluation of gene expression. The CFX Connect$^{TM}$ real-time PCR detection system (Bio-Rad, Watford, UK) was used for real-time quantitative PCR (RT-qPCR) with the following thermal cycling conditions: polymerase activation period at 95 °C for 10 min, followed by 40× PCR cycles of denaturation at 95 °C for 10 s and annealing/extension at 60 °C for 1 min.

**Small RNA expression analysis.** Small RNA expression analysis was performed on RNA samples used for gene expression analysis by Genewiz Inc., South Plainfield, NJ, USA.

**Statistical analysis.** All graphing and statistical analysis was performed using Prism software (Version 7.0a, GraphPad, USA). Quantitative measures and errors are presented as mean ± standard error of the mean (SEM) unless otherwise stated. Where data sets were large enough, the Shapiro–Wilk test for normality was performed and, if established, parametric tests applied. When two matched or unmatched groups were to be compared, paired or unpaired two-tailed t-tests were used respectively, with Welch's correction applied in cases where the standard deviation between the groups could not be assumed to be equal. In circumstances where three or more groups were to be compared, or the effects of multiple factors were to be determined within two or more groups, one- or two-way analysis of variance (ANOVA) tests were performed respectively. If significant statistical interaction was confirmed between factors, appropriate post-hoc tests were applied which accounted for multiple comparisons. In all cases, p values < 0.05 were considered statistically significant.

**Reporting summary**. Further information on research design is available in the Nature Research Reporting Summary linked to this article.

## Data availability

Source data are available via the Oxford research Archive: https://doi.org/10.5287/bodleian:MJ5GmD0Xv .Source data are provided with this paper.

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

## Acknowledgements

The authors would like to acknowledge the generous support of the Medical Research Council (UK) and Fight for Sight (UK) for providing funding for this work.

## Author contributions

H.O.O., M.E.M., A.R.B. and R.E.M. conceived the experiments. H.O.O., M.E.M. and C.M.F.C. carried out the experiments. H.O.O. drafted the paper. M.E.M., C.M.F.C. and R.E.M. edited the paper. All authors contributed to data analysis and approved the final paper.

## Competing interests

Authors on this paper (H.O.O., R.E.M., M.E.M., A.R.B.) are named inventors on two patents covering key intellectual property arising from this material, namely the AAV vector design (H.O.O., R.E.M., M.E.M., A.R.B.), and generic artificial mirtron-transgene constructs (H.O.O. & R.E.M.). The applicant for both patents is the University of Oxford. Patent international application numbers: PCT/GB2019/053036 and PCT/GB2019/053037. Patent international publications numbers: WO 2020/084318 A1 and WO 2020/084319 A1, both published 30/04/2020. C.M.F.C. declares no competing interests.
