## [Peer Review File · Nature Communications]

REVIEWER COMMENTS

Reviewer #1 (Remarks to the Author):

This paper describes the development of AAV delivered mirtrons (microRNAs processed by the spliceosome) for the treatment of autosomal dominant retinitis pigmentosa caused by mutations in the *RHO* gene, which codes for the rhodopsin. Though Curtis and colleagues have published similar experiments for SCA7 mutations (cited by Orleans et al. as reference 28), that work was purely in cell culture, and the present paper represents the first use of mirtrons in an animal model of disease. The paper is well written and the experiments are performed with appropriate controls. The statistical analysis was also appropriate. The results support the authors' conclusions, and, while the protection of the retina by these viral vectors was not overwhelming, this paper marks an advance in this technology that will certainly be improved by future experiments.

There are a few minor points and corrections that could improve this submission:

1. line 98: "Seven candidate mirtrons (M1-M7) were designed to target both mouse and human rhodopsin": Trying to target both genes must have severely limited the targetable sites in rhodopsin, given the difference in codon usage between mouse and human.
2. line 123: The off-target effects of miRNAs are associated with seed sequence (residues 2-7 of the mature miRNA) on the 3' UTR of gene expressed in the retina. To be thorough, they should have screened for non-specific targets using this criterion. Since that could result in hundreds of potential mRNAs to evaluate, that analysis should probably wait until they have selected a candidate mirtron for clinical translation.
3. line 196 and Figure 3d: There is substantial variation in the ERG response. Does this variation correspond to the extent of the retina that was transduced by the AAV vector?
4. line 223: Why was mirtron M5H used in the mouse experiments? Is that not the human specific mirtron?
5. line 297: What exactly was measured as the photoreceptor layer (PRL)? What landmarks were employed? Though the methods section was admirably complete, methods for SD-OCT were not provided. A reference to earlier work would do.
6. line 453: A reference is missing.

Reviewer #2 (Remarks to the Author):

Orlans et al., a novel mirtron knockdown/replacement gene therapy for the mutation-independent treatment of rhodopsin (RHO)-related autosomal dominant retinitis pigmentosa (ADRP), and have demonstrated efficacy in a relevant mammalian model. Splicing and potency of rhodopsin-targeting candidate mirtrons were initially determined and a mirtron-resistant codon-modified version of the human rhodopsin coding sequence was validated in vitro. These elements were then combined within an optimised expression cassette under the control of the rod cell-specific rhodopsin promoter, packaged within a single adeno associated virus (AAV) and delivered subretinally in a RhoP23H knock-in mouse model of ADRP. This treatment resulted in significant mouse-to-human rhodopsin RNA replacement and was associated with a slowing in the observed rate of retinal degeneration. This is a very good study which was performed very thoroughly. I recommend publication after my minor comments are addressed.

1. Fig. 6 shows that high dose had better knockdown and replacement, but the low dose had better protection. This was a little confusing can the authors please explain this discrepancy.
2. The data suggests moderate benefit. I think what will help is a comparison between other methods. Have other therapies been tested and what is the strength of the existing therapy is unclear.
3. An AAV strategy can only be a one-time use and it seems the therapeutic benefit is lost with time. Can this strategy be repeated without causing toxicity?

4. Some references in the method section have been jumbled.
5. Overall, this is an excellent study and I support its publication.

Reviewer #3 (Remarks to the Author):

Orlans et al report on a mirtron-based gene knockdown/replacement approach for autosomal dominant retinitis pigmentosa (ADRP). This approach provides an interesting and potentially good solution to the problem of autosomal dominant disease with mutational heterogeneity in that they are knocking down endogenous rhodopsin using a specialized microRNA called a mirtron and simultaneously delivering the wildtype gene modified to be resistant to mirtron targeting via an AAV-vector that houses both Rho and the mirtron expression cassette. A similar approach using mirtrons has been taken for knockdown/gene replacement in spinocerebellar ataxia 7 (Curtis et al., 2017). The general approach of knockdown/gene replacement has been shown by others to be effective in treating ADRP by targeting and replacing rhodopsin in a canine model (Cideciyan et al., 2018). In the current report the authors have modified the typical shRNA knockdown strategy to utilize mirtrons, with the potential advantage of producing miRNAs in a manner that won't overburden the natural endogenous pathway and can be expressed in a cell-type specific manner by virtue of the driving promoter, though it is not clear that this was necessarily problematic in the results reported in Cideciyan et al., 2018. Overall, the results are clear and the study overall is well-done and carefully controlled. The advance is not a major leap forward but is important for developing an optimal knockdown/replacement approach in ADRP. The manuscript is written in a way that may make it inaccessible to the non-expert reader and some of the statements are factually inaccurate as described in more detail below.

The lead sentence of the abstract seems out of place and could be moved to later in the paragraph. Also, the authors could add a sentence in the abstract to better introduce the problem that they are solving. For example, explain the nature of the ADRP mutation and the need to both eliminate expression of the gene encoding the toxic version and maintain or supplement the expression of the wt rhodopsin. For example, an abbreviated version of lines 51-59 would be good. Likewise, in the introduction, a few lines about ADRP and the disease mechanism and therapeutic goal should be added.

I don't understand this concept described on line 145: "Placement of mirtrons within the CDS for knockdown/replacement gene therapy may adversely affect transgene expression²⁸, whilst placement within the 3'-UTR would likely result in nonsense-mediated decay²⁹. We therefore explored placing the most effective mirtrons within the 5'-UTR of the eGFP transgene in vitro (Fig. 2a)." I think what the authors are suggesting is that an unspliced mirtron (retained intron) may cause nonsense-mediated decay. However, this is unlikely to occur in the context of a single intron and in the untranslated region.

The statements surrounding splicing mechanisms are over-simplified and the writing/interpretation would benefit from guidance from an expert. For example, the statement on line 359 is misplaced "Taken together, these data would suggest that the mammalian splicing machinery may recognize a range of donor motifs that differ significantly from the GURAGU consensus in artificial mirtrons." The recognition and selection of donor splice sites is a central question in the splicing field that is still not well understood. It is well known that many sites that don't match the consensus sequence are very efficiently spliced. Please remove or modify this statement to capture better the meaning of this result, which is not surprising or unexpected.

In addition, this statement on line 357: "This region of the intron is critical for donor splice site recognition by the small nuclear RNA U6 component of the spliceosome^{38,39}." Is incomplete as the +4 to +6 positions of the splice site also base-pair with U1 snRNA.

Likewise, the statement on line 350 is imprecise:

"These designs, with the exception of M4, fulfilled the criterion of Seow et al. that artificial mirtrons should commence with a GURRR motif to encourage splicing. M4 had a C at position 352 +5 but nevertheless demonstrated reasonable splicing." The term "to encourage" splicing and "reasonable splicing" are non-scientific and should be avoided. The GURAGU sequence is the mammalian consensus sequence of donor splice sites and is required for splicing. This is different than an enhancer splicing sequence that might be included to "encourage" splicing.

The authors cite Seow et al as a reference for their design rationale. It would be very useful for the authors to include a diagram of their mirtrons similar to that shown in Fig. 2D of Seow et al., so that the reader can see the entire mirtron sequence with the donor and acceptor splice sites.

The amount of miRNA produced from the mirtron should be quantitated, in particular in the in vivo study. The analysis and study is incomplete without a thorough analysis of all the components of the system.

Figure 5 a,b and c,d why were two different types of graphs used? The authors should present the individual mice point in 5b and d similar to 5a and c.

Some of the references did not get incorporated (lines 453, 597, 599).

Mirtron-mediated RNA knockdown/replacement therapy for the treatment of dominant retinitis pigmentosa: Response to reviewers

Harry O. Orlans, Michelle E. McClements, Alun R. Barnard, Cristina Martinez-Fernandez de la Camara, Robert E. MacLaren

Reviewer #1 (Remarks to the Author):

This paper describes the development of AAV delivered mirtrons (microRNAs processed by the spliceosome) for the treatment of autosomal dominant retinitis pigmentosa caused by mutations in the RHO gene, which codes for the rhodopsin. Though Curtis and colleagues have published similar experiments for SCA7 mutations (cited by Orlans et al. as reference 28), that work was purely in cell culture, and the present paper represents the first use of mirtrons in an animal model of disease. The paper is well written and the experiments are performed with appropriate controls. The statistical analysis was also appropriate. The results support the authors' conclusions, and, while the protection of the retina by these viral vectors was not overwhelming, this paper marks an advance in this technology that will certainly be improved by future experiments.

We would like to thank the reviewer for his/her positive comments.

There are a few minor points and corrections that could improve this submission:

1. line 98: "Seven candidate mirtrons (M1-M7) were designed to target both mouse and human rhodopsin": Trying to target both genes must have severely limited the targetable sites in rhodopsin, given the difference in codon usage between mouse and human.

As explained in lines 137-138, potential targets were chosen in areas of homology between the transcripts of the two species so that a vector validated in mouse may be directly applied in a future human clinical trial. The reviewer is quite correct that this strategy had the drawback of limiting the number of potential target sites. We have now included a sentence in the discussion to highlight this point (lines 500-502).

2. line 123: The off-target effects of miRNAs are associated with seed sequence (residues 2-7 of the mature miRNA) on the 3' UTR of gene expressed in the retina. To be thorough, they should have screened for non-specific targets using this criterion. Since that could result in hundreds of potential mRNAs to evaluate, that analysis should probably wait until they have selected a candidate mirtron for clinical translation.

The reviewer is correct in this regard. We have now included a passage to acknowledge this in the section of the supplementary material related to the assessment of off-target effects (Section S1, lines 36-39, and supplementary reference 2).

3. line 196 and Figure 3d: There is substantial variation in the ERG response. Does this variation correspond to the extent of the retina that was transduced by the AAV vector?

We did not specifically correlate extent of transduction observed in retinal sections with ERG signal in our study as not all eyes were processed for immunohistochemistry. We did however keep records of all subretinal injections performed. Of the four treated eyes which gave the lowest signal, two had documented surgical complications: one had substantial transscleral reflux

of the vector observed at the time of injection resulting in a shallow retinal detachment, whilst in another significant intraoperative subretinal haemorrhage was noted. All animals showing high ERG signal after treatment had documented uncomplicated good quality injections. In two of the treated eyes with no documented intraoperative complications, a low ERG signal was recorded. We attribute this to possible trans-scleral reflux of the vector after completion of the injection. We have included this discussion in our revised manuscript (lines 245 – 252).

4. line 223: Why was mirtron M5H used in the mouse experiments? Is that not the human specific mirtron?

Two mirtrons were included in the final construct so that delivery of multiple independent mirtrons could be demonstrated in vivo. The human rather than mouse version of M5 was chosen so that the vector may be appropriate for subsequent use in patients. The M5 mirtron would thus not be expected to be active in mice in the in vivo section of this study. We have now added a passage to the discussion section of the revised manuscript to clarify this point (496-499).

5. line 297: What exactly was measured as the photoreceptor layer (PRL)? What landmarks were employed? Though the methods section was admirably complete, methods for SD-OCT were not provided. A reference to earlier work would do.

This has now been defined with a corresponding reference (lines 365-366 and reference 37).

6. line 453: A reference is missing.

Reference now included.

Reviewer #2 (Remarks to the Author):

Orlans et al., a novel mirtron knockdown/replacement gene therapy for the mutation-independent treatment of rhodopsin (RHO)-related autosomal dominant retinitis pigmentosa (ADRP), and have demonstrated efficacy in a relevant mammalian model. Splicing and potency of rhodopsin-targeting candidate mirtrons were initially determined and a mirtron-resistant codon-modified version of the human rhodopsin coding sequence was validated in vitro. These elements were then combined within an optimised expression cassette under the control of the rod cell-specific rhodopsin promoter, packaged within a single adeno associated virus (AAV) and delivered subretinally in a RhoP23H knock-in mouse model of ADRP. This treatment resulted in significant mouse-to-human rhodopsin RNA replacement and was associated with a slowing in the observed rate of retinal degeneration. This is a very good study which was performed very thoroughly. I recommend publication after my minor comments are addressed.

We would like to thank the reviewer for his/her positive comments.

1. Fig. 6 shows that high dose had better knockdown and replacement, but the low dose had better protection. This was a little confusing can the authors please explain this discrepancy.

We explore the possible causes of this discrepancy in the discussion (lines 491-496). To clarify this further, we have now expanded this section and included a reference to recent work supporting this hypothesis (reference 23).

2. The data suggests moderate benefit. I think what will help is a comparison between other methods. Have other therapies been tested and what is the strength of the existing therapy is unclear.

Other RNA replacement therapies have been tested in animal models of rhodopsin-related RP including a transgenic P23H mouse model (Mao et al, 2012), a transgenic P347S model (Millington-Ward et al 2011), and a naturally-occurring dog model (Cideciyan et al 2018). Given the variety of animal models involved which all have very different rates of retinal degeneration, it is difficult to directly compare the efficacies of the various vectors described with that of our own. This point has been included in the revised discussion (lines 409-411).

The mirtron approach has several theoretical advantages over the methods outlined in existing studies which we have outlined in the introduction and discussion. Whether or not these are borne out in practice remains to be determined. We have now included this point in our revised discussion (lines 428-430).

To our knowledge, the only other study exploring the utility of the knockdown/replacement approach in the heterozygous P23H knock-in mouse model (described in Sakami et al 2011) is that of Tsai et al 2018: Clustered Regularly Interspaced Short Palindromic Repeats-Based Genome Surgery for the Treatment of Autosomal Dominant Retinitis Pigmentosa. Ophthalmology. The magnitude of our rescue effect as measured by ERG (Fig. 6d-f) is comparable to that shown by Tsai et al in their study (see Fig 4e of Tsai et al 2018). We have now included this paper in our revised discussion section (lines 411-416).

Other therapeutic strategies have been attempted in the P23H knock-in mouse model. We have explored short wavelength light filtration as a potential therapeutic in this model (see Orleans et al 2019). The current therapy does not suffer from many of the limitations associated with the light filtration approach which include alteration of colour perception and reduced colour discrimination, potential for negative cosmetic effects and reliance on long-term compliance with the treatment.

3. An AAV strategy can only be a one-time use and it seems the therapeutic benefit is lost with time. Can this strategy be repeated without causing toxicity?

We do not know at present whether repeat treatment with our vector would increase the therapeutic effect or induce retinal toxicity. We acknowledge that this would be an interesting avenue of future enquiry. All AAV vectors are approved for single use and whilst there is clearly ongoing degeneration in cells that have not been transduced at the correct dose, the current evidence in humans suggests that at least some of the effects are long lasting.

4. Some references in the method section have been jumbled.

This issue has been resolved in the revised manuscript.

5. Overall, this is an excellent study and I support its publication.

Again, we would like to thank the reviewer for his/her strong support for the publication of our research in Nature Communications.

Reviewer #3 (Remarks to the Author):

Orlans et al report on a mirtron-based gene knockdown/replacement approach for autosomal dominant retinitis pigmentosa (ADRP). This approach provides an interesting and potentially good solution to the problem of autosomal dominant disease with mutational heterogeneity in that they are knocking down endogenous rhodopsin using a specialized microRNA called a mirtron and simultaneously delivering the wildtype gene modified to be resistant to mirtron targeting via an AAV-vector that houses both Rho and the mirtron expression cassette. A similar approach using mirtrons has been taken for knockdown/gene replacement in spinocerebellar ataxia 7 (Curtis et al., 2017). The general approach of knockdown/gene replacement has been shown by others to be effective in treating ADRP by targeting and replacing rhodopsin in a canine model (Cideciyan et al., 2018). In the current report the authors have modified the typical shRNA knockdown strategy to utilize mirtrons, with the potential advantage of producing miRNAs in a manner that won't overburden the natural endogenous pathway and can be expressed in a cell-type specific manner by virtue of the driving promoter, though it is not clear that this was necessarily problematic in the results reported in Cideciyan et al., 2018. Overall, the results are clear and the study overall is well-done and carefully controlled.

We would like to thank the reviewer for their positive comments.

The advance is not a major leap forward but is important for developing an optimal knockdown/replacement approach in ADRP. The manuscript is written in a way that may make it inaccessible to the non-expert reader and some of the statements are factually inaccurate as described in more detail below.

We hope that our revised manuscript is clearer and we have addressed the inaccuracies highlighted below.

The lead sentence of the abstract seems out of place and could be moved to later in the paragraph. Also, the authors could add a sentence in the abstract to better introduce the problem that they are solving. For example, explain the nature of the ADRP mutation and the need to both eliminate expression of the gene encoding the toxic version and maintain or supplement the expression of the wt rhodopsin. For example, an abbreviated version of lines 51-59 would be good. Likewise, in the introduction, a few lines about ADRP and the disease mechanism and therapeutic goal should be added.

We have modified the abstract to better introduce the problem as suggested whilst still adhering to the 150 word limit imposed by the journal. The introduction has also been rewritten to take on board these suggestions.

I don't understand this concept described on line 145: "Placement of mirtrons within the CDS for knockdown/replacement gene therapy may adversely affect transgene expression²⁸, whilst placement within the 3'-UTR would likely result in nonsense-mediated decay²⁹. We therefore explored placing the most effective mirtrons within the 5'-UTR of the eGFP transgene in vitro (Fig. 2a)." I think what the authors are suggesting is that an unspliced mirtron (retained intron) may cause nonsense-mediated decay. However, this is unlikely to occur in the context of a single intron and in the untranslated region.

We considered three locations to place our mirtrons in relation to the transgene within our expression cassette: within the coding sequence (CDS), within the 3'-UTR and within the 5'-UTR.

A retained intron in the coding sequence (CDS) is likely to cause a frame-shift, premature stop codon and trigger nonsense-mediated decay (NMD). We know from our in vitro experiments that a proportion of mirtrons do not splice, or splice incorrectly as a result of alternative donor/acceptor sites (see Fig. 1d). Indeed, reduced transgene expression as a result of mirtron inclusion in the CDS was shown in the study of Curtis et al 2017. We therefore chose not to embed mirtrons within the RHO CDS in our expression cassette.

The reason we chose not to locate mirtrons in the 3'-UTR of our transgene is that this too might in theory trigger NMD. The mechanism of NMD is suggested to rely upon detection of exon-exon junctional complexes (EJCs). These protein complexes are deposited 20-24 nt upstream of 5' splice sites after splicing and are usually removed by ribosomes during translation. Introns downstream of a stop codon will not have their EJCs removed in this way (as the ribosome has already separated from the transcript). The presence of retained EJCs on mRNA is thought to be the trigger for NMD. A premature stop codon induces NMD of the transcript in this manner. Please see Maquat et al 2004 Nat Rev Mol Cell Biol, and Baker & Parker 2004 Curr. Opin Cell Biol for detailed reviews on this subject. On this basis, we reasoned that placing mirtrons downstream of the stop codon of our transgene (i.e. in the 3'-UTR) might trigger degradation of the whole transcript. Since placement within the CDS or within the 3'-UTR might ultimately trigger NMD, we chose to design our expression cassette with mirtrons located within the 5'-UTR of the transgene. We are sorry that we did not make this argument clear in our original submission and have included a more in depth explanation in our revised manuscript (lines 185-192).

The statements surrounding splicing mechanisms are over-simplified and the writing/interpretation would benefit from guidance from an expert. For example, the statement on line 359 is misplaced "Taken together, these data would suggest that the mammalian splicing machinery may recognize a range of donor motifs that differ significantly from the GURAGU consensus in artificial mirtrons." The recognition and selection of donor splice sites is a central question in the splicing field that is still not well understood. It is well known that many sites that don't match the consensus sequence are very efficiently spliced. Please remove or modify this statement to capture better the meaning of this result, which is not surprising or unexpected. In addition, this statement on line 357: "This region of the intron is critical for donor splice site recognition by the small nuclear RNA U6 component of the spliceosome^{38,39}." Is incomplete as the +4 to+6 positions of the splice site also base-pair with U1 snRNA.

Likewise, the statement on line 350 is imprecise:

"These designs, with the exception of M4, fulfilled the criterion of Seow et al. that artificial mirtrons should commence with a GURRR motif to encourage splicing. M4 had a C at position 352 +5 but nevertheless demonstrated reasonable splicing." The term "to encourage" splicing and "reasonable splicing" are non-scientific and should be avoided. The GURAGU sequence is the mammalian consensus sequence of donor splice sites and is required for splicing. This is different than an enhancer splicing sequence that might be included to "encourage" splicing.

The revised manuscript has been modified to incorporate these suggested changes.

The authors cite Seow et al as a reference for their design rationale. It would be very useful for the authors to include a diagram of their mirtrons similar to that shown in Fig. 2D of Seow et al., so that the reader can see the entire mirtron sequence with the donor and acceptor splice sites.

Methods Fig. 2 has now been modified to include full mirtron sequence diagrams as requested.

The amount of miRNA produced from the mirtron should be quantitated, in particular in the in vivo study. The analysis and study is incomplete without a thorough analysis of all the components of the system.

We have shown in great detail the development of a functional mirtron gene therapy vector, efficient liberation of mirtrons from an AAV-derived transgene in vivo and first presentation of efficacy of mirtron-induced knockdown in vivo, as well as rescue of a relevant disease model. On this basis we do not believe that our study is incomplete as suggested. Whilst we agree that quantification of mirtron-derived miRNA would be of interest, we feel that this analysis would be more appropriate as part of follow-on work we are currently undertaking in which we are evaluating and comparing different vectors with various combinations of RHO-targeting mirtrons.

Figure 5 a,b and c,d why were two different types of graphs used? The authors should present the individual mice point in 5b and d similar to 5a and c.

The figure has been modified as suggested.

Some of the references did not get incorporated (lines 453, 597, 599).

This issue has been resolved in the revised manuscript.

REVIEWER COMMENTS

Reviewer #1 (Remarks to the Author):

The authors have revised the paper and addressed the concerns raised in the previous review.

Alfred Lewin

Reviewer #2 (Remarks to the Author):

I recommend publication

Reviewer #3 (Remarks to the Author):

I appreciate the thoughtful attention to my comments and concerns. I cannot agree with your dismissal of the importance of measuring miRNA levels. The amount of mature miRNA is one of the most important parts of this system and without showing that you are achieving reasonable maturation (not just splicing to release the pre-miRNA) you do not have a direct demonstration of the mechanism of action nor a good assessment of the relationship between miRNA abundance and target knockdown. This point was one of the first things that I was looking for when I read the manuscript and I suspect that to a broader audience interested in the approach, the absence of this information will be disappointed and also potentially questionable in its validity.

Mirtron-mediated RNA knockdown/replacement therapy for the treatment of dominant retinitis pigmentosa: Response to reviewers

Harry O. Orlans, Michelle E. McClements, Alun R. Barnard, Cristina Martinez-Fernandez de la Camara, Robert E. MacLaren

Reviewer #1 (Remarks to the Author):

The authors have revised the paper and addressed the concerns raised in the previous review.

Alfred Lewin

We would like to thank Professor Lewin for his support for our manuscript.

Reviewer #2 (Remarks to the Author):

I recommend publication

We would like to thank the reviewer for their support for our manuscript.

Reviewer #3 (Remarks to the Author):

I appreciate the thoughtful attention to my comments and concerns. I cannot agree with your dismissal of the importance of measuring miRNA levels. The amount of mature miRNA is one of the most important parts of this system and without showing that you are achieving reasonable maturation (not just splicing to release the pre-miRNA) you do not have a direct demonstration of the mechanism of action nor a good assessment of the relationship between miRNA abundance and target knockdown. This point was one of the first things that I was looking for when I read the manuscript and I suspect that to a broader audience interested in the approach, the absence of this information will be disappointed and also potentially questionable in its validity.

We have now addressed this outstanding question with an in depth analysis of micro RNAs from injected samples (see lines 279-328). We would like to thank the reviewer for suggesting this further work which we feel adds substantial weight to the study.

REVIEWERS' COMMENTS

Reviewer #3 (Remarks to the Author):

The authors have addressed my concerns by including additional data.